# POLYNOMIAL GRAPH CONVOLUTIONAL NETWORKS

## ABSTRACT

Graph Convolutional Neural Networks (GCNs) exploit convolution operators, based on some neighborhood aggregating scheme, to compute representations of graphs. The most common convolution operators only exploit local topological information. To consider wider topological receptive fields, the mainstream approach is to non-linearly stack multiple Graph Convolutional (GC) layers. In this way, however, interactions among GC parameters at different levels pose a bias on the flow of topological information. In this paper, we propose a different strategy, considering a single graph convolution layer that independently exploits neighbouring nodes at different topological distances, generating decoupled representations for each of them. These representations are then processed by subsequent readout layers. We implement this strategy introducing the Polynomial Graph Convolution (PGC) layer, that we prove being more expressive than the most common convolution operators and their linear stacking. Our contribution is not limited to the definition of a convolution operator with a larger receptive field, but we prove both theoretically and experimentally that the common way multiple non-linear graph convolutions are stacked limits the neural network expressiveness. Specifically, we show that a Graph Neural Network architecture with a single PGC layer achieves state of the art performance on many commonly adopted graph classification benchmarks.

## 1 INTRODUCTION

In the last few years, the definition of machine learning methods, particularly neural networks, for graph-structured input has been gaining increasing attention in literature (Defferrard et al., 2016; Errica et al., 2020). In particular, Graph Convolutional Networks (GCNs), based on the definition of a convolution operator in the graph domain, are relatively fast to compute and have shown good predictive performance. Graph Convolutions (GC) are generally based on a neighborhood aggregation scheme (Gilmer et al., 2017) considering, for each node, only its direct neighbors. Stacking multiple GC layers, the size of the receptive field of deeper filters increases (resembling standard convolutional networks). However, stacking too many GC layers may be detrimental on the network ability to represent meaningful topological information (Li et al., 2018) due to a too high Laplacian smoothing. Moreover, in this way interactions among GC parameters at different layers pose a bias on the flow of topological information. For these reasons, several convolution operators have been defined in literature, differing from one another in the considered aggregation scheme. We argue that the performance of GC networks could benefit by increasing the size of the receptive fields, but since with existing GC architectures this effect can only be obtained by stacking more GC layers, the increased difficulty in training and the limitation of expressiveness given by the stacking of many local layers ends up hurting their predictive capabilities.

Consequently, the performances of existing GCNs are strongly dependent on the specific architecture. Therefore, existing graph neural network performances are limited by *(i)* the necessity to select an appropriate convolution operator, and *(ii)* the limitation of expressiveness caused by large receptive fields being possible only stacking many local layers.

In this paper, we tackle both the issues following a different strategy. We propose the Polynomial Graph Convolution (PGC) layer that independently considers neighbouring nodes at different topological distances (i.e. arbitrarily large receptive fields). The PGC layer faces the problem of selecting a suitable convolution operator being able to represent many existing convolutions in literature, and being more expressive than most of them. As for the second issue a PGC layer, directly considering larger receptive fields, can represent a richer set of functions compared to the *linear stacking* of two or more Graph Convolution layers, i.e. it is more expressive. Moreover, the linear PGC design allows

to consider large receptive fields without incurring in typical issues related to training deep networks. We developed the Polynomial Graph Convolutional Network (PGCN), an architecture that exploits the PGC layer to perform graph classification tasks. We empirically evaluate the proposed PGCN on eight commonly adopted graph classification benchmarks. We compare the proposed method to several state-of-the-art GCNs, consistently achieving higher or comparable predictive performances. Differently from other works in literature, the contribution of this paper is to show that the common approach of stacking multiple GC layers may not provide an optimal exploitation of topological information because of the strong coupling of the depth of the network with the size of the topological receptive fields. In our proposal, the depth of the PGCN is decoupled from the receptive field size, allowing to build deep GNNs avoiding the oversmoothing problem.

## 2    NOTATION

We use italic letters to refer to variables, bold lowercase to refer to vectors, and bold uppercase letters to refer to matrices. The elements of a matrix $\mathbf{A}$ are referred to as $a_{ij}$ (and similarly for vectors). We use uppercase letters to refer to sets or tuples. Let $G = (V, E, \mathbf{X})$ be a graph, where $V = \{v_0, \ldots, v_{n-1}\}$ denotes the set of vertices (or nodes) of the graph, $E \subseteq V \times V$ is the set of edges, and $\mathbf{X} \in \mathbb{R}^{n \times s}$ is a multivariate signal on the graph nodes with the $i$-th row representing the attributes of $v_i$. We define $\mathbf{A} \in \mathbb{R}^{n \times n}$ as the adjacency matrix of the graph, with elements $a_{ij} = 1 \iff (v_i, v_j) \in E$. With $\mathcal{N}(v)$ we denote the set of nodes adjacent to node $v$. Let also $\mathbf{D} \in \mathbb{R}^{n \times n}$ be the diagonal degree matrix where $d_{ii} = \sum_j a_{ij}$, and $\mathbf{L}$ the normalized graph laplacian defined by $\mathbf{L} = \mathbf{I} - \mathbf{D}^{-\frac{1}{2}} \mathbf{A} \mathbf{D}^{-\frac{1}{2}}$, where $\mathbf{I}$ is the identity matrix.

With $GConv_\theta(\mathbf{x}_v, G)$ we denote a graph convolution with set of parameters $\theta$. A GCN with $k$ levels of convolutions is denoted as $GConv_{\theta_k}(\ldots GConv_{\theta_1}(\mathbf{x}_v, G) \ldots, G)$. For a discussion about the most common GCNs we refer to Appendix A. We indicate with $\hat{\mathbf{X}}$ the input representation fed to a layer, where $\hat{\mathbf{X}} = \mathbf{X}$ if we are considering the first layer of the graph convolutional network, or $\hat{\mathbf{X}} = \mathbf{H}^{(i-1)}$ if considering the $i$-th graph convolution layer.

## 3    POLYNOMIAL GRAPH CONVOLUTION (PGC)

In this section, we introduce the Polynomial Graph Convolution (PGC), able to simultaneously and directly consider all topological receptive fields up to $k - hops$, just like the ones that are obtained by the graph convolutional layers in a stack of size $k$. PGC, however, does not incur in the typical limitation related to the complex interaction among the parameters of the GC layers. Actually, we show that PGC is *more* expressive than the most common convolution operators. Moreover, we prove that a single PGC convolution of order $k$ is capable of implementing $k$ linearly stacked layers of convolutions proposed in the literature, providing also *additional* functions that *cannot* be realized by the stack. Thus, the PGC layer extracts topological information from the input graph decoupling in an effective way the depth of the network from the size of the receptive field. Its combination with deep MLPs allows to obtain deep graph neural networks that can overcome the common oversmoothing problem of current architectures. The basic idea underpinning the definition of PGC is to consider the case in which the graph convolution can be expressed as a polynomial of the powers of a transformation $\mathcal{T}$ of the adjacency matrix. This definition is very general, and thus it incorporates many existing graph convolutions as special cases. Given a graph $G = (V, E, \mathbf{X})$ with adjacency matrix $\mathbf{A}$, the Polynomial Graph Convolution (PGC) layer of degree $k$, transformation $\mathcal{T}$ of $\mathbf{A}$, and size $m$, is defined as

$$PGConv_{k,\mathcal{T},m}(\mathbf{X}, \mathbf{A}) = \mathbf{R}_{k,\mathcal{T}}\mathbf{W}, \tag{1}$$

where $\mathcal{T}(\mathbf{A}) \in \mathbb{R}^{n \times n}$, $\mathbf{R}_{k,\mathcal{T}} \in \mathbb{R}^{n \times s*(k+1)}$, $\mathbf{R}_{k,\mathcal{T}} = [\mathbf{X}, \mathcal{T}(\mathbf{A})\mathbf{X}, \mathcal{T}(\mathbf{A})^2\mathbf{X}, .., \mathcal{T}(\mathbf{A})^k\mathbf{X}]$, and $\mathbf{W} \in \mathbb{R}^{s*(k+1) \times m}$ is a learnable weight matrix. For the sake of presentation, we will consider $\mathbf{W}$ as composed of blocks: $\mathbf{W} = [\mathbf{W}_0, \ldots, \mathbf{W}_k]^\top$, with $\mathbf{W}_j \in \mathbb{R}^{s \times m}$. In the following, we show that PGC is very expressive, able to implement commonly used convolutions as special cases.

### 3.1 Graph Convolutions in literature as PGC instantiations

The PGC layer in equation 1 is designed to be a generalization of the linear stacking of some of the most common spatially localized graph convolutions. The idea is that spatially localized convolutions aggregate over neighbors (the message passing phase) using a transformation of the adjacency matrix (e.g. a normalized graph Laplacian). We provide in this section a formal proof, as a theoretical contribution of this paper, that linearly stacked convolutions can be rewritten as polynomials of powers of the transformed adjacency matrix.

We start showing how common graph convolution operators can be defined as particular instances of a single PGC layer (in most cases with $k = 1$). Then, we prove that linearly stacking any two PGC layers produces a convolution that can be written as a single PGC layer as well.

**Spectral:** The Spectral convolution (Defferrard et al., 2016) can be considered the application of Fourier transform to graphs. It is obtained via Chebyshev polynomials of the Laplacian matrix. A layer of Spectral convolutions of order $k^\star$ can be implemented by a single PGC layer instantiating $\mathcal{T}(A)$ to be the graph Laplacian matrix (or one of its normalized versions), setting the PGC $k$ value to $k^\star$, and setting the weight matrix to encode the constraints given by the Chebyshev polynomials. For instance, we can get the output $\mathbf{H}$ of a Spectral convolution layer with $k^\star = 3$ by the following PGC:

$$\mathbf{H} = [\mathbf{X}, \mathbf{LX}, \mathbf{L}^2, \mathbf{L}^3 \mathbf{X}]\mathbf{W}, \text{ where } \mathbf{W} = \begin{bmatrix} \mathbf{W}_0 - \mathbf{W}_2 \\ \mathbf{W}_1 - 3\mathbf{W}_3 \\ 2\mathbf{W}_2 \\ 4\mathbf{W}_3 \end{bmatrix}, \ \mathbf{W}_i \in \mathbb{R}^{s \times m}. \tag{2}$$

**GCN:** The Graph Convolution (Kipf & Welling, 2017) (GCN) is a simplification of the spectral convolution. The authors propose to fix the order $k^\star = 1$ of the Chebyshev spectral convolution to obtain a linear first order filter for each graph convolutional layer in a neural network. By setting $k = 1$ and $\mathcal{T}(A) = \tilde{\mathbf{D}}^{-\frac{1}{2}} \tilde{\mathbf{A}} \tilde{\mathbf{D}}^{-\frac{1}{2}} \in \mathbb{R}^{n \times n}$, with $\tilde{\mathbf{A}} = \mathbf{A} + \mathbf{I}$ and $\tilde{d}_{ii} = \sum_j \tilde{a}_{ij}$, we obtain the following equivalent equation:

$$\mathbf{H} = [\mathbf{X}, \tilde{\mathbf{D}}^{-\frac{1}{2}} \tilde{\mathbf{A}} \tilde{\mathbf{D}}^{-\frac{1}{2}} \mathbf{X}]\mathbf{W}, \text{ where } \mathbf{W} = \begin{bmatrix} \mathbf{0} \\ \mathbf{W}_1 \end{bmatrix}, \tag{3}$$

where $\mathbf{0}$ is a $s \times m$ matrix with all entries equal to zero and $\mathbf{W}_1 \in \mathbb{R}^{s \times m}$ is the weight matrix of GCN. Note that the GCN does not consider a node differently from its neighbors, thus in this case there is no contribution from the first component of $\mathbf{R}_{k, \mathcal{T}}$.

***GraphConv:*** In Morris et al. (2019) a powerful graph convolution has been proposed, that is inspired by the Weisfeiler-Lehman graph invariant. In this case, $\mathcal{T}(A) = A$ (the identity function), and $k$ is again set to 1. A single *GraphConv* layer can be written as:

$$\mathbf{H} = [\mathbf{X}, \mathbf{AX}]\mathbf{W}, \text{ where } \mathbf{W} = \begin{bmatrix} \mathbf{W}_0 \\ \mathbf{W}_1 \end{bmatrix}, \text{ and } \mathbf{W}_0, \mathbf{W}_1 \in \mathbb{R}^{s \times m}. \tag{4}$$

**GIN:** The Graph Isomorphism Network (GIN) convolution was defined in Xu et al. (2019) as: $\mathbf{H}' = MLP((1 + \epsilon)\hat{\mathbf{X}} + \mathbf{A}\hat{\mathbf{X}})$. Technically, this is a composition of a convolution (that is a linear operator) with a multi-layer perceptron. Let us thus decompose the $MLP()$ as $f \circ g$, where $g$ is an affine projection via weight matrix $\mathbf{W}$, and $f$ incorporates the element-wise non-linearity, and the other layers of the MLP. We can thus isolate the GIN graph convolution component and define it as a specific PGC istantiation. We let $k = 1$ and $\mathcal{T}()$ the identity function as before. A single GIN layer can then be obtained as:

$$\mathbf{H} = [\mathbf{X}, \mathbf{AX}]\mathbf{W}, \text{ where } \mathbf{W} = \begin{bmatrix} (1 + \epsilon)\mathbf{W}_1 \\ \mathbf{W}_1 \end{bmatrix}. \tag{5}$$

Note that, differently from *GraphConv*, in this case the blocks of the matrix $\mathbf{W}$ are tied. Figure 1 in Appendix B depicts the expressivity of different graph convolution operators in terms of the respective constraints on the weight matrix $\mathbf{W}$. The comparison is made easy by the definition of the different graph convolution layers as instances of PGC layers. Actually, it is easy to see from eqs. (3)-(5) that *GraphConv* is more expressive than GCN and GIN.

### 3.2 Linearly Stacked Graph Convolutions as PGC instantiations

In the previous section, we have shown that common graph convolutions can be expressed as particular instantiations of a PGC layer. In this section, we show that a single PGC layer can model the linear stacking of any number of PGC layers (using the same transformation $\mathcal{T}$). Thus, a single PGC layer can model all the functions computed by arbitrarily many linearly stacked graph convolution layers defined in the previous section. We then show that a PGC layer includes also *additional* functions compared to the stacking of simpler PGC layers, which makes it more expressive.

**Theorem 1.** *Let us consider two linearly stacked PGC layers using the same transformation $\mathcal{T}$. The resulting linear Graph Convolutional network can be expressed by a single PGC layer.*

Due to space limitations, the proof is reported in appendix C. Here it is important to know that the proof of Theorem 1 tells us that a single PGC of order $k$ can represent the linear stacking of any $q$ ($\mathcal{T}$-compatible) convolutions such that $k = \sum_{i=1}^{q} d_i$, where $d_i$ is the degree of convolution at level $i$. We will now show that a single PGC layer can represent also other functions, i.e. it is more general than the stacking of existing convolutions. Let us consider, for the sake of simplicity, the stacking of 2 PGC layers with $k = 1$ (that are equivalent to *GraphConv* layers, see eq. equation 4), each with parameters $\mathbf{W}^{(i)} = [\mathbf{W}_0^{(i)}, \mathbf{W}_1^{(i)}]^\top$, $i = 1, 2$, $\mathbf{W}_0^{(1)}, \mathbf{W}_1^{(1)} \in \mathbb{R}^{s \times m_1}$, $\mathbf{W}_0^{(2)}, \mathbf{W}_1^{(2)} \in \mathbb{R}^{m_1 \times m_2}$. The same reasoning can be applied to any other convolution among the ones presented in Section 3.1. We can explicitly write the equations computing the hidden representations:

$$\mathbf{H}^{(1)} = \mathbf{X}\mathbf{W}_0^{(1)} + \mathbf{A}\mathbf{X}\mathbf{W}_1^{(1)}, \tag{6}$$

$$\mathbf{H}^{(2)} = \mathbf{H}^{(1)}\mathbf{W}_0^{(2)} + \mathbf{A}\mathbf{H}^{(1)}\mathbf{W}_1^{(2)} \tag{7}$$
$$= \mathbf{X}\mathbf{W}_0^{(1)}\mathbf{W}_0^{(2)} + \mathbf{A}\mathbf{X}(\mathbf{W}_1^{(1)}\mathbf{W}_0^{(2)} + \mathbf{W}_0^{(1)}\mathbf{W}_1^{(2)}) + \mathbf{A}^2\mathbf{X}\mathbf{W}_1^{(1)}\mathbf{W}_1^{(2)}.$$

A single PGC layer can implement this second order convolution as:

$$\mathbf{H}^{(2)} = [\mathbf{X}, \mathbf{A}\mathbf{X}, \mathbf{A}^2\mathbf{X}] \begin{bmatrix} \mathbf{W}_0^{(1)}\mathbf{W}_0^{(2)} \\ \mathbf{W}_1^{(1)}\mathbf{W}_0^{(2)} + \mathbf{W}_0^{(1)}\mathbf{W}_1^{(2)} \\ \mathbf{W}_1^{(1)}\mathbf{W}_1^{(2)} \end{bmatrix}. \tag{8}$$

Let us compare it with a PGC layer that corresponds to the same 2-layer architecture but that has no constraints on the weight matrix, i.e.:

$$\mathbf{H}^{(2)} = [\mathbf{X}, \mathbf{A}\mathbf{X}, \mathbf{A}^2\mathbf{X}] \begin{bmatrix} \mathbf{W}_0 \\ \mathbf{W}_1 \\ \mathbf{W}_2 \end{bmatrix}, \ \mathbf{W}_i \in \mathbb{R}^{s \times m_2}, \ i = 0, 1, 2. \tag{9}$$

Even though it is not obvious at a first glance, equation 8 is more constrained than equation 9, i.e. there are some values of $\mathbf{W}_0, \mathbf{W}_1, \mathbf{W}_2$ in equation 9 that cannot be obtained for any $\mathbf{W}^{(1)} = [\mathbf{W}_0^{(1)}, \mathbf{W}_1^{(1)}]^\top$ and $\mathbf{W}^{(2)} = [\mathbf{W}_0^{(2)}, \mathbf{W}_1^{(2)}]^\top$ in equation 8, as proven by the following theorem.

**Theorem 2.** *A PGC layer with $k = 2$ is more general than two stacked PGC layers with $k = 1$ with the same number of hidden units $m$.*

We refer the reader to Appendix C for the proof. Notice that the *GraphConv* layer is equivalent to a PGC layer with $k = 1$ (if no constraints on $\mathbf{W}$ are considered, see later in this section). Since the *GraphConv* is, in turn, more general than GCN and GIN, the above theorem holds also for those graph convolutions. Moreover, Theorem 2 trivially implies that a linear stack of $q$ PGC layers with $k = 1$ is less expressive than a single PGC layer with $k = q$.

If we now consider that in many GCN architectures it is typical, and useful, to concatenate the output of all convolution layers before aggregating the node representations, then it is not difficult to see that such concatenation can be obtained by making wider the weight matrix of PGC. Let us thus consider a network that generates a hidden representation that is the concatenation of the different representations computed on each layer, i.e. $\mathbf{H} = [\mathbf{H}^{(1)}, \mathbf{H}^{(2)}] \in \mathbb{R}^{s \times m}$, $m = m_1 + m_2$. We can represent a 2-layer *GraphConv* network as a single PGC layer as:

$$\mathbf{H} = [\mathbf{X}, \mathbf{A}\mathbf{X}, \mathbf{A}^2\mathbf{X}] \begin{bmatrix} \mathbf{W}_0^{(1)} & \mathbf{W}_0^{(1)}\mathbf{W}_0^{(2)} \\ \mathbf{W}_1^{(1)} & \mathbf{W}_1^{(1)}\mathbf{W}_0^{(2)} + \mathbf{W}_0^{(1)}\mathbf{W}_1^{(2)} \\ \mathbf{0} & \mathbf{W}_1^{(1)}\mathbf{W}_1^{(2)} \end{bmatrix}. \tag{10}$$

More in general, if we consider $k$ *GraphConv* convolutional layers (see eq. (4)), each with parameters $\mathbf{W}^{(i)} = [\mathbf{W}_0^{(i)}, \mathbf{W}_1^{(i)}]^\top, i = 1, \ldots, k$, $\mathbf{W}_0^{(i)}, \mathbf{W}_1^{(i)} \in \mathbb{R}^{m_{i-1} \times m_i}$, $m_0 = s$, $m = \sum_{j=1}^k$, the weight matrix $\mathbf{W} \in \mathbb{R}^{s \cdot (k+1) \times m}$ can be defined as follows:

$$\begin{bmatrix} F_{0,1}(\mathbf{W}^{(1)}) & F_{0,2}(\mathbf{W}^{(1)}, \mathbf{W}^{(2)}) & F_{0,3}(\mathbf{W}^{(1)}, \mathbf{W}^{(2)}, \mathbf{W}^{(3)}) & \ldots \\ F_{1,1}(\mathbf{W}^{(1)}) & F_{1,2}(\mathbf{W}^{(1)}, \mathbf{W}^{(2)}) & F_{1,3}(\mathbf{W}^{(1)}, \mathbf{W}^{(2)}, \mathbf{W}^{(3)}) & \ldots \\ \mathbf{0} & F_{2,2}(\mathbf{W}^{(1)}, \mathbf{W}^{(2)}) & F_{2,3}(\mathbf{W}^{(1)}, \mathbf{W}^{(2)}, \mathbf{W}^{(3)}) & \ldots \\ \mathbf{0} & \mathbf{0} & F_{3,3}(\mathbf{W}^{(1)}, \mathbf{W}^{(2)}, \mathbf{W}^{(3)}) & \ldots \\ \ldots & \ldots & \ldots & \ldots \end{bmatrix}, \tag{11}$$

where $F_{i,j}(), \ i, j \in \{0, \ldots, k\}, \ i \leq j$, are defined as

$$F_{i,j}(\mathbf{W}^{(1)}, \ldots, \mathbf{W}^{(j)}) = \sum_{\substack{(z_1, .., z_j) \in \{0,1\}^j \\ s.t. \ \sum_{q=1}^j z_q = i}} \prod_{s=1}^j \mathbf{W}_{z_s}^{(s)}.$$

We can now generalize this formulation by concatenating the output of $k + 1$ PGC convolutions of degree ranging from 0 up to $k$. This gives rise to the following definitions:

$$\mathbf{W} = \begin{bmatrix} \mathbf{W}_{0,0} & \mathbf{W}_{0,1} & \mathbf{W}_{0,2} & \ldots & \mathbf{W}_{0,k} \\ \mathbf{0} & \mathbf{W}_{1,1} & \mathbf{W}_{1,2} & \ldots & \mathbf{W}_{1,k} \\ \mathbf{0} & \mathbf{0} & \mathbf{W}_{2,2} & \ldots & \mathbf{W}_{2,k} \\ \vdots & \vdots & \vdots & \ddots & \vdots \\ \mathbf{0} & \mathbf{0} & \mathbf{0} & \ldots & \mathbf{W}_{k,k} \end{bmatrix}, \ \mathbf{H} = \begin{bmatrix} (\mathbf{X}\mathbf{W}_{0,0})^\top \\ (\mathbf{X}\mathbf{W}_{0,1} + \mathcal{T}(\mathbf{A})\mathbf{X}\mathbf{W}_{1,1})^\top \\ \vdots \\ (\mathbf{X}\mathbf{W}_{0,k} + \cdots + \mathcal{T}(\mathbf{A})^k \mathbf{X}\mathbf{W}_{k,k})^\top \end{bmatrix}^\top \tag{12}$$

where we do not put constraints among matrices $\mathbf{W}_{i,j} \in \mathbb{R}^{s \times m_j}$, $m = \sum_{j=0}^k m_j$, which are considered mutually independent. Note that as a consequence of Theorem 2, the network defined in equation 12 is more expressive than the one obtained concatenating different *GraphConv* layers as defined in equation 11. It can also be noted that the same network can actually be seen as a single PGC layer of order $k + 1$ with a constraint on the weight matrix (i.e., to be an upper triangular block matrix). Of course, any weights sharing policy can be easily implemented, e.g. by imposing $\forall j \ \mathbf{W}_{i,j} = \mathbf{W}_i$, which corresponds to the concatenation of the representations obtained at level $i$ by a single stack of convolutions. In addition to reduce the number of free parameters, this weights sharing policy allows the reduction of the computational burden since the representation at level $i$ is obtained by summing to the representation of level $i - 1$ the contribution of matrix $\mathbf{W}_i$, i.e. $\mathbf{A}^i \mathbf{X} \mathbf{W}_i$

### 3.3 COMPUTATIONAL COMPLEXITY

As detailed in the previous discussion, the *degree* $k$ of a PGC layer controls the size of its considered receptive field. In terms of the number of parameters, fixing the node attribute size $s$ and the size $m$ of the hidden representation, the number of parameters of the PGC is $O(s \cdot k \cdot m)$, i.e. it grows linearly in $k$. Thus, the number of parameters of a PGC layer is of the same order of magnitude compared to $k$ stacked graph convolution layers based on message passing (Gilmer et al., 2017) (i.e. *GraphConv*, GIN and GCN, presented in Section 3.1).

If we consider the number of required matrix multiplications, compared to message passing GC networks, in our case it is possible to pre-compute the terms $\mathcal{T}(\mathbf{A})^i \mathbf{X}$ before the start of training, making the computational cost of the convolution calculation cheaper compared to message passing. In Appendix E, we report an example that makes evident the significant improvement that can be gained in training time with respect to message passing.

## 4 POLYNOMIAL GRAPH CONVOLUTIONAL NETWORK (PGCN)

In this section, we present a neural architecture that exploits the PGC layer to perform graph classification tasks. Note that, differently from other GCN architectures, in our architecture (exploiting the PGC layer) the depth of the network is completely decoupled from the size $k$ of the receptive field. The initial stage of the model consists of a first PGC layer with $k = 1$. The role of this first layer is to develop an initial node embedding to help the subsequent PGC layer to fully exploit its power. In fact,

in bioinfromatics datasets where node labels $\mathbf{X}$ are one-hot encoded, all matrices $\mathbf{X}, \mathbf{A}\mathbf{X}, \ldots, \mathbf{A}^k\mathbf{X}$ are very sparse, which we observed, in preliminary experiments, influences in a negative way learning. Table 4 in Appendix F compares the sparsity of the PGC representation using the original one-hot encoded labels against their embedding obtained with the first PGC layer. The analysis shows that using this first layer the network can work on significantly denser representations of the nodes. Note that this first stage of the model does not significantly bound the PGC-layer expressiveness. A dense input for the PGC layer could have been obtained by using an embedding layer that is not a graph convolutional operator. However, this choice would have made difficult to compare our results with other state-of-the-art models in Section 6, since the same input transformation could have been applied to other models as well, making unclear the contribution of the PGC layer to the performance improvement. This is why we decided to use a PGC layer with $k = 1$ (equivalent to a *GraphConv*) to compute the node embedding, making the results fully comparable since we are using only graph convolutions in our PGCN. For what concerns the datasets that do not have the nodes' label (like the social networks datasets), using the PGC layer with $k = 1$ allows to create a label for each node that will be used by the subsequent larger PGC layer to compute richer node's representations. After this first PGC layer, a PGC layer as described in eq. equation 12 of degree $k$ is applied. In order to reduce the number of hyperparameters, we adopted the same number $\frac{m}{k+1}$ of columns (i.e., hidden units) for matrices $\mathbf{W}_{i,j}$, i.e. $\mathbf{W}_{i,j} \in \mathbb{R}^{s \times \frac{m}{k+1}}$. A graph-level representation $\mathbf{s} \in \mathbb{R}^{m*3}$ based on the PGC layer output $\mathbf{H}$ is obtained by an aggregation layer that exploits three different aggregation strategies over the whole set of nodes $V$, $j = 1, \ldots, m$:

$$s_j^{avg} = avg(\{h_v^{(j)}|v \in V\}), \ s_j^{max} = max(\{h_v^{(j)}|v \in V\}), \ s_j^{sum} = sum(\{h_v^{(j)}|v \in V\}),$$
$$\mathbf{s} = [s_1^{avg}, s_1^{max}, s_1^{sum}, \ldots, s_m^{avg}, s_m^{max}, s_m^{sum}]^\top.$$

The readout part of the model is composed of $q$ dense feed-forward layers, where we consider $q$ and the number of neurons per layer as hyper-parameters. Each one of these layers uses the *ReLU* activation function, and is defined as $\mathbf{y}_j = ReLu(\mathbf{W}_j^{readout}\mathbf{y}_{j-1} + \mathbf{b}^{readout})$, $j = 0, \ldots, q$, where $\mathbf{y}_0 = \mathbf{s}$. Finally, the output layer of the PGCN for a $c$-class classification task is defined as: $\mathbf{o} = LogSoftmax(\mathbf{W}^{out}\mathbf{y}_q + \mathbf{b}^{out})$.

## 5 COMPARISON VS MULTI-SCALE GCN ARCHITECTURES IN LITERATURE

Some recent works in literature exploit the idea of extending graph convolution layers to increase the receptive field size. In general, the majority of these models, that concatenate polynomial powers of the adjacency matrix A, are designed to perform node classification, while the proposed PGCN is developed to perform graph classification. In this regard, we want to point out that the novelty introduced in this paper is not limited to a novel GC-layer, but the proposed PGCN is a complete architecture to perform graph classification. Atwood & Towsley (2016) proposed a method that exploits the power series of the probability transition matrix, that is multiplied (using Hadamard product) by the inputs. The method differs from the PGCN even in terms of how the activation is computed and even because the activation computed for each exponentiation is summed, instead been concatenated. Similarly in Defferrard et al. (2016) the model exploits the Chebyshev polynomials, and, differently from PGCN sums them over k. This architectural choice makes the proposed method less general than the PGCN. Indeed, as showed in Section 3.1, the model proposed in (Defferrard et al., 2016) is an instance of the PGC.

In (Xu et al., 2018) the authors proposed to modify the common aggregation layer in such a way that, for each node, the model aggregates all the intermediate representations computed in the previous GC-layers. In this work, differently from PGCN, the model exploits the message passing method introducing a bias in the flow of the topological information. Note that, as proven in Theorem 2, a PGC layer of degree k is not equivalent to concatenate the output of k stacked GC layers, even though the PGC layer can also implement this particular architecture.

Another interesting approach is proposed in (Tran et al., 2018), where the authors consider larger receptive fields compared to standard graph convolutions. However, they focus on a single convolution definition (using just the adjacency matrix) and consider shortest paths (differently from PGCN that exploits matrix exponentiations, i.e. random walks). In terms of expressiveness, it is complex to compare methods that exploit matrix exponentiations with methods based on the shortest paths. However, it is interesting to notice that, thanks to the very general structure of the PGC layer, it is

easy to modify the PGC definition in order to use the shortest paths instead of the adjacency matrix transformation exponentiations. In particular, we plan to explore this option as the future development of the PGCN.

Wu *et al.* introduce a simplification of the graph convolution operator, dubbed Simple Graph Convolution (SGC) (Wu et al., 2019). The model proposed is based on the idea that perhaps the nonlinear operator introduced by GCNs is not essential, and basically, the authors propose to stack several linear GC operators. In Theorem2 we prove that staking k GC layers is less expressive than using a single PGC layer of degree k. Therefore, we can conclude that the PGC Layer is a generalization of the SGC.

In (Liao et al., 2019) the authors construct a deep graph convolutional network, exploiting particular localized polynomial filters based on the Lanczos algorithm, which leverages multi-scale information. This convolution can be easily implemented by a PGC layer. In (Chen et al., 2019) the authors propose to replace the neighbor aggregation function with graph augmented features. These graph augmented features combine node degree features and multi-scale graph propagated features. Basically, the proposed model concatenates the node degree with the power series of the normalized adjacency matrix. Note that the graph augmented features differ from $\mathbf{R}_{k,\mathcal{T}}$, used in the PGC layer. Another difference with respect to the PGCN resides on the subsequent part of the model. Indeed, instead of projecting the multi-scale features layer using a structured weights matrix, the model proposed in (Chen et al., 2019) aggregates the graph augmented features of each vertex and project each of these subsets by using an MLP. The model readout then sums the obtained results over all vertices and projects it using another MLP.

Luan et al. (2019) introduced two deep GCNs that rely on Krylov blocks. The first one exploits a GC layer, named snowball, that concatenates multi-scale features incrementally, resulting in a densely-connected graph network. The architecture stacks several layers and exploits nonlinear activation functions. Both these aspects make the gradient flow more complex compared to the PGCN. The second model, called Truncated Krylov, concatenates multi-scale features in each layer. In this model, differently from PCGN, the weights matrix of each layer has no structure, thus topological features from all levels are mixed together. A similar approach is proposed in (Rossi et al., 2020). Rossi *et. al* proposed an alternative method, named SIGN, to scale GNN to a very large graph. This method uses as a building block the set of exponentiations of linear diffusion operators. In this building block, every exponentiation of the diffusion operator is linearly projected by a learnable matrix. Moreover, differently from the PGC layer, a nonlinear function is applied on the concatenation of the diffusion operators making the gradient flow more complex compared to the PGCN.

Very recently Liu et al. (2020) proposed a model dubbed Deep Adaptive Graph Neural Network, to learn node representations by adaptively incorporating information from large receptive fields. Differently from PGCN, first, the model exploits an MLP network for node feature transformation. Then it constructs a multi-scale representation leveraging on the computed nodes features transformation and the exponentiation of the adjacency matrix. This representation is obtained by stacking the various adjacency matrix exponentiations (thus obtaining a 3-dimensional tensor). Similarly to (Luan et al., 2019) also in this case the model projects the obtained multi-scale representation using weights matrix that has no structure, obtaining that the topological features from all levels are mixed together. Moreover, this projection uses also a (trainable) retainment scores. These scores measure how much information of the corresponding representations derived by different propagation layers should be retained to generate the final representation for each node in order to adaptively balance the information from local and global neighborhoods. Obviously, that makes the gradient flow more complex compared to the PGCN, and also impact the computational complexity.

## 6 EXPERIMENTAL SETUP AND RESULTS

In this section, we introduce our model set up, the adopted datasets, the baselines models, and the hyper-parameters selection strategy. We then report and discuss the results obtained by the PGCN. For implementation details please refer to Appendix G.

**Datasets.** We empirically validated the proposed PGCN on five commonly adopted graph classification benchmarks modeling bioinformatics problems: PTC (Helma et al., 2001), NCI1 (Wale et al., 2008), PROTEINS, (Borgwardt et al., 2005), D&D (Dobson & Doig, 2003) and ENZYMES (Borg-

| Model \ Dataset | PTC | NCI1 | PROTEINS | D&D | ENZYMES | COLLAB | IMDB-B | IMDB-M |
|---|---|---|---|---|---|---|---|---|
| PSCN[1] | 60.00 ±4.82 | 76.34 ±1.68 | 75.00 ±2.51 | 76.27 ±2.64 | - - | 72.60 ±2.15 | 71.00 ±2.29 | 45.23 ±2.84 |
| FGCNN[2] | 58.82 ±1.80 | 81.50 ±0.39 | 74.57 ±0.80 | 77.47 ±0.86 | - - | - - | - - | - - |
| DGCNN[2] | 57.14 ±2.19 | 72.97 ±0.87 | 73.96 ±0.41 | 78.09 ±0.72 | - - | - - | - - | - - |
| DGCNN[3] | - - | 76.4 ±1.7 | 72.9 ±3.5 | 76.6 ±4.3 | 38.9 ±5.7 | 57.4 ±1.9 | 53.3 ±5.0 | 38.6 ±2.2 |
| GIN[3] | - - | 80.0 ±1.4 | 73.3 ±4.0 | 75.3 ±2.9 | 59.6 ±4.5 | **75.9** ±**1.9** | 66.8 ±3.9 | 42.2 ±4.6 |
| DIFFPOOL[3] | - - | 76.9 ±1.9 | 73.7 ±3.5 | 75.0 ±3.5 | 59.5 ±5.6 | 67.7 ±1.9 | 68.3 ±6.1 | 45.1 ±3.2 |
| GraphSAGE[3] | - - | 76.0 ±1.8 | 73.0 ±4.5 | 72.9 ±2.0 | 58.2 ±6.0 | 71.6 ±1.5 | 69.9 ±4.6 | 47.2 ±3.6 |
| Baseline[3] | - - | 69.8 ±2.2 | **75.8** ±**3.7** | 78.4 ±4.5 | 65.2 ±6.4 | 55.0 ±1.9 | 50.7 ±2.4 | 36.1 ±3.0 |
| PGCN | **60.50** ±**0.67** | **82.04** ±**0.26** | 75.31 ±0.31 | **79.45** ±**0.29** | **70.5** ±1.77 | 74.1 ±1.69 | **72.60** ±**3.80** | **47.39** ±**3.51** |

Table 1: Accuracy comparison between PGCNN and several *state-of-the-art* models on graph classification task. [1](Niepert et al., 2016) , [2](Navarin et al., 2020), [3](Errica et al., 2020).

wardt et al., 2005). Moreover, we also evaluated the PGCN on 3 large graph social datasets: COLLAB, IMDB-B, IMDB-M (Yanardag & Vishwanathan, 2015). We report more details in Appendix D.

**Baselines and Hyper-parameter selection.** We compare PGCN versus several GNN architectures, that achieved state-of-the-art results on the used datasets. Specifically, we considered PSCN (Niepert et al., 2016), Funnel GCNN (FGCNN) model (Navarin et al., 2020), DGCNN (Zhang et al., 2018), GIN (Xu et al., 2019), DIFFPOOL (Ying et al., 2018) and GraphSage (Hamilton et al., 2017). Note that these graph classification models exploit the convolutions presented in Section 3.1. From (Errica et al., 2020) we report also the results of a baseline models that is structure-agnostic.
The results were obtained by performing 5 runs of 10-fold cross-validation. The hyper-parameters of the model (number of hidden units, learning rate, weight decay, $k$, $q$) were selected using a grid search, where the explored sets of values were changed based on the considered dataset. Other details about validation are reported in Appendix I.
The results reported in Xu et al. (2019); Chen et al. (2019); Ying et al. (2018) are not considered in our comparison since the model selection strategy is different from the one we adopted and this makes the results not comparable. The importance of the validation strategy is discussed in Errica et al. (2020), where results of a fair comparison among the considered baseline models are reported. For the sake of completeness, we also report (and compare) in Appendix J the results obtained by evaluating the PGCN method with the validation policy used in Xu et al. (2019); Chen et al. (2019); Ying et al. (2018).

## 6.1 RESULTS AND DISCUSSION

The results reported in Table 1 show that the PGCN achieves higher results in all (except one) considered datasets compared to competing methods. In particular, on NCI1 and ENZYMES the proposed method outperforms state-of-the-art results. In fact, in both cases, the performances of PGCN and the best competing method are more than one standard deviation apart. Even for PTC, D&D, PROTEINS, IMDB-B and IMDB-M datasets PGCN shows a slight improvement over the results of FGCNN and DGCNN models. Furthermore, the results of PGCN in Bioinformatics datasets achieves a significant lower standard deviation (evaluated over the 5 runs of 10-fold cross-validation). For what concerns the COLLAB datasets, PGCN obtained the second higher result in the considered state-of-the-art methods. Note however that the difference with respect to the first one (GIN) is within the standard deviation.

| Dataset \ k | 3 | 4 | 5 | 6 |
|---|---|---|---|---|
| **PTC** | **74.0** $\pm$**2.81** | 68.86 $\pm$2.44 | 69.14 $\pm$2.33 | 69.43 $\pm$1.78 |
| **NCI1** | 82.68 $\pm$0.22 | 83.16 $\pm$0.70 | **84.40** $\pm$**0.14** | 84.04 $\pm$0.83 |
| **PROTEINS** | **79.20** $\pm$**0.81** | 78.48 $\pm$0.56 | 78.48 $\pm$1.04 | 78.84 $\pm$0.80 |
| **D&D** | 81.95 $\pm$1.19 | 82.03 $\pm$1.06 | 81.69 $\pm$0.68 | **82.54** $\pm$**1.09** |
| **ENZYMES** | 77.50 $\pm$1.27 | 76.83 $\pm$0.85 | **78.17** $\pm$**0.68** | 77.18 $\pm$0.92 |

| Dataset \ k | 3 | 5 | 7 | 9 |
|---|---|---|---|---|
| **COLLAB** | 76.94 $\pm$1.47 | 76.32 $\pm$1.45 | **76.96** $\pm$**2.14** | 76.72 $\pm$1.60 |
| **IMDB-B** | 76.70 $\pm$3.48 | 76.60 $\pm$3.84 | 76.38 $\pm$2.90 | **77.88** $\pm$**3.11** |
| **IMDB-M** | 52.37 $\pm$3.02 | 52.7 $\pm$2.84 | 52.41 $\pm$2.85 | **52.97** $\pm$**2.67** |

Table 2: PGCN accuracy comparison on the validation set of the datasets for different values of $k$.

**Significativity of our results.** To understand if the improvements reported in Table 1 are significant or can be attributed to random chance, we conducted the two-tailed Wilcoxon Signed-Ranks test between our proposed PGCN and competing methods. This test considers all the results for the different datasets at the same time. According to this test, PGCN performs significantly better ($p$-value $< 0.05$) than PSCN, DGCNN[3], GIN, DIFFPOOL and GraphSAGE. As for FGCNN and DGCNN[2], four datasets are not enough to conduct the test.

**Impact of receptive field size on PGCN.** Most of the proposed GCN architectures in literature generally stack 4 or fewer GCs layers. The proposed PGC layer allows us to represent a linear version of these architectures by using a single layer with an even higher depth ($k$), without incurring in problems related to the flow of the topological information. Different values of $k$ have been tested to study how much the capability of the model to represent increased topological information helps to obtain better results. The results of these experiments are reported in Table 2. The accuracy results in this table are referred to the *validation* sets, since the choice of $k$ is part of the model selection procedure. We decided to take into account a range of $k$ values between 3 and 6 for bioinformatics datasets, and between 3 to 9 for social networks datasets. The results show that it is crucial to select an appropriate value for $k$. Several factors influence how much depth is needed. It is important to take into account that the various datasets used for the experiments refer to different tasks. The quantity and the type of topological information required (or useful) to solve the task highly influences the choice of $k$. Moreover, also the input dimensions and the number of graphs contained in a dataset play an important role. In fact, using higher values of $k$ increases the number columns of the $\mathbf{R}_{k,\mathcal{T}}$ matrix (and therefore the number of parameters embedded in $\mathbf{W}$), making the training of the model more difficult. It is interesting to notice that in many cases our method exploits a larger receptive field (i.e. a higher *degree*) compared to the competing models. Note that the datasets where better results are obtained with $k = 3$ (PTC and PROTEINS) contain a limited amount of training samples, thus deeper models tend to overfit arguably for the limited amount of training data.

## 7 Conclusions and Future Works

In this paper, we analyze some of the most common convolution operators evaluating their expressiveness. Our study shows that their linear composition can be defined as instances of a more general Polynomial Graph Convolution operator with a higher expressiveness. We defined an architecture exploiting a single PGC layer to generate a decoupled representation for each neighbor node at a different topological distance. This strategy allows us to avoid the bias on the flow of topological information introduced by stacking multiple graph convolution layers. We empirically validated the proposed Polynomial Graph Convolutional Network on five commonly adopted graph classification benchmarks. The results show that the proposed model outperforms competing methods in almost all the considered datasets, showing also a more stable behavior.

In the future, we plan to study the possibility to introduce an *attention mechanism* by learning a transformation $\mathcal{T}$ that can adapt to the input. Furthermore, we will explore whether adopting our PGC operator as a large random projection can allow to develop a novel model for learning on graph domains.

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

## A GRAPH NEURAL NETWORKS

A Graph Neural Network (GNN) is a neural network model that exploits the structure of the graph and the information embedded in feature vectors of each node to learn a classifier or regressor on a graph domain. Due to the success in image processing, convolutional-based neural networks have become one of the main architectures (ConvGNN) applied to graph processing. The typical structure of a ConvGNN comprises a first part of processing where convolutional layers are used in order to learn a representation $\mathbf{h}_v \in \mathbb{R}^m$ for each vertex $v \in V$. These representations are then *combined* to get a representation of the whole graph, so that a standard feed-forward (deep) neural network can be used to process it. Convolutional layers are important since they define *how* the (local) topological information is mixed with the information attached to involved nodes, and *what* is the information to pass over to the subsequent computational layers. Because of that, several convolution operators for graphs have recently been proposed (Defferrard et al., 2016; Kipf & Welling, 2017; Morris et al., 2019; Xu et al., 2019).

The first definition of neural network for graphs has been proposed by Sperduti & Starita (1997).More recently, Micheli (2009) proposed the Neural Network for Graphs (NN4G), exploiting an idea that has been re-branded later as *graph convolution*, and Scarselli et al. (2008) defined a recurrent neural network for graphs.In the last few years, several models inspired by the graph convolution have been proposed. Many recent works defining graph convolutional networks (GCNs) extend the NN4G formulation (Micheli, 2009), for instance the Graph Convolutional Network (GCN) (Kipf & Welling, 2017) is based on a linear first-order filter based on the normalized graph Laplacian for each graph convolutional layer in a neural network. SGC (Wu et al., 2019) proposes a fast way to compute the result of several linearly stacked GCNs. Note that SGC considers just the GCN convolution, while our proposed PGC is more expressive than any number of linearly stacked graph convolutions among the ones presented in Section 3.1 of the main paper. DGCNN (Zhang et al., 2018) adopts a graph convolution very similar to GCN (Kipf & Welling, 2017) (a slightly different propagation scheme for vertices' representations is defined, based on the random-walk graph Laplacian). While GCN is focused on node classification, DGCNN is suited for graph classification since it incorporates the readout.

A more straightforward approach in defining convolutions on graphs is PATCHY-SAN (PSCN) (Niepert et al., 2016). This approach is inspired by how convolutions are defined over images. It consists in selecting a fixed number of vertices from each graph and exploiting a canonical ordering on graph vertices. For each vertex, it defines a fixed-size neighborhood, exploiting the same vertex ordering. It requires the vertices of each input graph to be in a canonical ordering, which is as complex as the graph isomorphism problem (no polynomial-time algorithm is known).

Another interesting proposal for the convolution over the node neighborhood is GraphSage Hamilton et al. (2017), which proposes to perform an aggregation over the neighborhoods by using sum, mean or max-pooling operators, and then perform a linear projection in order to update the node representation. In addition to that, the proposed approach exploits a particular neighbors sampling scheme. GIN (Xu et al., 2019) is an extension of GraphSage that avoids the limitation introduced by using sum, mean or max-pooling by using a more expressive aggregation function on multi-sets. DiffPool (Ying et al., 2018) is an end-to-end architecture that combines a differentiable graph encoder with its polling mechanism. Indeed, the method learns an adaptive pooling strategy to collapse nodes on the basis of a supervised criterion.

The Funnel GCNN (FGCNN) model (Navarin et al., 2020) aims to enhance the gradient propagation using a simple aggregation function and LeakyReLU activation functions. Hinging on the similarity of the adopted graph convolutional operator, that is the *GraphConv*, to the way feature space representations by Weisfeiler-Lehman (WL) Subtree Kernel (Shervashidze et al., 2011) are generated, it introduces a loss term for the output of each convolutional layer to guide the network to reconstruct the corresponding explicit WL features. Moreover, the number of filters used at each convolutional layer is based on a measure of the WL kernel complexity.

## B EXPRESSIVENESS OF COMMONLY USED GRAPH CONVOLUTIONS

Thanks to the possibility to express commonly used graph convolutions as instances of PGC, and from the discussion in Section 3 of the paper, it is easy to characterize the expressiveness of commonly

used graph convolutions. In Figure 1 we represent the inclusion relationships among the sets of functions which can be implemented by GCN, GIN, GraphConv, Spectral.

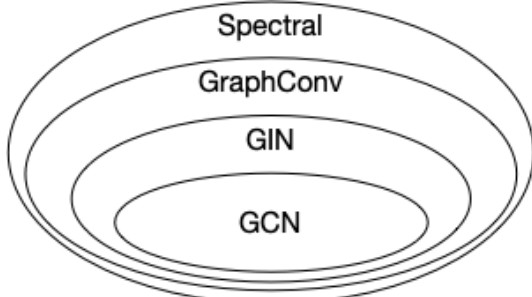

Figure 1: Expressiveness of commonly used graph convolution operators. Each ellipse represents the set of functions that can be implemented by a single graph convolution operator.

## C   Proofs of Theorems

**Theorem 1.** *Let us consider two linearly stacked PGC layers using the same transformation $\mathcal{T}$. The resulting linear Graph Convolutional network can be expressed by a single PGC layer.*

*Proof.* With no loss of generality, let the first PGC being of degree $k_1$, while the second stacked PGC of degree $k_2$, i.e.

$$\mathbf{H}^{(1)} = [\mathbf{X}, \ldots, \mathcal{T}(\mathbf{A})^{k_1}\mathbf{X}]\mathbf{W}^{(1)},$$

$$\mathbf{H}^{(2)} = [\mathbf{H}^{(1)}, \ldots, \mathcal{T}(\mathbf{A})^{k_2}\mathbf{H}^{(1)}]\mathbf{W}^{(2)},$$

where

$$\mathbf{W}^{(i)} = \begin{bmatrix} \mathbf{W}_0^{(i)} \\ \mathbf{W}_1^{(i)} \\ \vdots \\ \mathbf{W}_{k_i}^{(i)} \end{bmatrix}, \ i = 1, 2.$$

By expanding $\mathbf{H}^{(1)}$ inside $\mathbf{H}^{(2)}$ equation, we get:

$$\begin{aligned} \mathbf{H}^{(2)} = &[[\mathbf{X}\mathbf{W}_0^{(1)} + \ldots + \mathcal{T}(\mathbf{A})^{k_1}\mathbf{X}\mathbf{W}_{k_1}^{(1)}], \ldots, \\ &\mathcal{T}(\mathbf{A})^{k_2}[\mathbf{X}\mathbf{W}_0^{(1)} + \ldots + \mathcal{T}(\mathbf{A})^{k_1}\mathbf{X}\mathbf{W}_{k_1}^{(1)}]]\mathbf{W}^{(2)} \\ = &[\mathbf{X}\mathbf{W}_0^{(1)}\mathbf{W}_0^{(2)} + \ldots + \mathcal{T}(\mathbf{A})^{k_1}\mathbf{X}\mathbf{W}_{k_1}^{(1)}\mathbf{W}_0^{(2)} + \\ &\ldots + \mathcal{T}(\mathbf{A})^{k_2}\mathbf{X}\mathbf{W}_0^{(1)}\mathbf{W}_{k_2}^{(2)} + \ldots + \\ &\mathcal{T}(\mathbf{A})^{k_1+k_2}\mathbf{X}\mathbf{W}_{k_1}^{(1)}\mathbf{W}_{k_2}^{(2)}]. \end{aligned} \tag{13}$$

In this case, by defining $D_1 = \{0, .., k_1\}$, $D_2 = \{0, .., k_2\}$, and auxiliary functions $F_i()$, $i = 0, .., k_1 + k_2$, defined as

$$F_i(\mathbf{W}^{(1)}, \mathbf{W}^{(2)}) = \sum_{\substack{(z_1,z_2)\in D_1\times D_2 \\ s.t. \ z_1+z_2=i}} \mathbf{W}_{z_1}^{(1)}\mathbf{W}_{z_2}^{(2)},$$

matrix $\mathbf{W}$ can be written as

$$\mathbf{W} = \begin{bmatrix} F_0(\mathbf{W}^{(1)}, \mathbf{W}^{(2)}) \\ F_1(\mathbf{W}^{(1)}, \mathbf{W}^{(2)}) \\ \cdots \\ F_{k_1+k_2}(\mathbf{W}^{(1)}, \mathbf{W}^{(2)}) \end{bmatrix},$$

and consequently the hidden representation becomes

$$\mathbf{H}^{(2)} = [\mathbf{X}, \mathcal{T}(\mathbf{A})\mathbf{X}, \ldots, \mathcal{T}(\mathbf{A})^{k_1}\mathbf{X}, \ldots, \mathcal{T}(\mathbf{A})^{k_1+k_2}\mathbf{X}]\mathbf{W},$$

which is the output of a PGC with $k = k_1 + k_2$. $\square$

**Theorem 2.** *A PGC layer with $k = 2$ is more general than two stacked PGC layers with $k = 1$ with the same transformation $\mathcal{T}$ and the same number of hidden units $m$.*

*Proof.* Since all PGCs use the same transformation $\mathcal{T}$, we can focus on the weights only. We prove our theorem providing a counterexample, i.e. we fix $m = 1$, $s = 1$ (the input dimension) and show an instantiation of the weight matrix $\mathbf{W} = [\mathbf{W}_0^\top, \mathbf{W}_1^\top, \mathbf{W}_2^\top]^\top$ of a PGC layer with $k = 2$ that cannot be expressed by the composition of two PGC layers with $k = 1$ (equivalent to *GraphConv*). Let us consider the simplest case in which $\mathbf{W}_0, \mathbf{W}_1, \mathbf{W}_2 \in \mathbb{R}$, i.e. they are $1 \times 1$ matrices. Let us now consider the case where $\mathbf{W}_0 = 5$, $\mathbf{W}_1 = 7$, $\mathbf{W}_2 = 3$. Let us also, for the sake of clarity, rename the $1 \times 1$ matrices of the two PGCs with $k = 1$ as: $\mathbf{W}_0^{(1)} \leftarrow a$, $\mathbf{W}_0^{(2)} \leftarrow b$, $\mathbf{W}_1^{(1)} \leftarrow c$, $\mathbf{W}_1^{(2)} \leftarrow d$. We get the following system of equations:

$$\begin{cases} ab = 5 \\ ad + cb = 7 \\ cd = 3 \end{cases} \quad \begin{cases} a = 5/b \\ d = 3/c \\ 15/cb + cb = 7 \end{cases} \quad \begin{cases} a = 5/b \\ d = 3/c \\ (cb)^2 - 7cb + 15 = 0 \end{cases} \tag{14}$$

where we assume $b$ and $c$ are different from zero (it is easy to see that either $b = 0$ or $c = 0$ would not lead to any solution). If we compute the $\Delta$ of the third equation (solving for $cb$), we get $\Delta = \sqrt{49 - 60} = i\sqrt{11}$, i.e. a complex number. Thus there is no real value for $cb$ that satisfies our system of equations. We thus conclude that there are no values that we can assign to the parameters of the PGCs with $k = 1$ that would lead to the considered PGC weight matrix. $\square$

Moreover, Theorem 2 implies the following corollary.

**Corollary 2.1.** *A linear stack of $q$ $\mathcal{T}$-compatible PGC layers with $k = 1$ is less expressive than a single $\mathcal{T}$-PGC layer with $k = q$.*

*Proof.* Since all PGCs use the same transformation $\mathcal{T}$, we can focus on the weights only. We prove the corollary by induction. The base case is provided by Theorem 2. Let us now prove the inductive case. Let us assume that a stack of $i$ PGC layers with $k = 1$ (with parameters set $\boldsymbol{\theta}_1^{(i)} = \{\mathbf{W}^{(j)} \in \mathbb{R}^{2s \times m}, \; j = 0, \ldots, i\}$) is less expressive than a single PGC layer with $k = i$ (with parameters set $\boldsymbol{\theta}_2^{(i)} = \mathbf{W} \in \mathbb{R}^{(i+1)s \times m}$), and let us prove the same result for $i + 1$. We can consider the set of functions that can be implemented by the stack of $i + 1$ PGC layers with $k = 1$ as the composition of two functions coming from two different sets: the first one $PGC_{k=1}^{(i)} = \{f \mid \exists \hat{\boldsymbol{\theta}}_1^{(i)} \; s.t. \; f \equiv PGC_{k=1}^{(i)}(\hat{\boldsymbol{\theta}}_1^{(i)})\}$, is the set of functions that can be computed stacking $i$ PGC layers with $k = 1$, and the second one $PGC_{k=1}^{(1)} = \{f \mid \exists \hat{\boldsymbol{\theta}}_1^{(1)} \; s.t. \; f \equiv PGC_{k=1}^{(1)}(\hat{\boldsymbol{\theta}}_1^{(1)})\}$, is the set of functions computed by a single PGC layer with $k = 1$. We can then characterize the set of functions $PGC_{k=1}^{(i+1)}$ as:

$$PGC_{k=1}^{(i+1)} = \{f \circ g \mid f \in PGC_{k=1}^{(i)}, g \in PGC_{k=1}^{(i)}\}.$$

From Theorem 1, we know that $PGC_{k=i+1}^{(1)} \supseteq \{f \circ g \mid f \in PGC_{k=i}^{(1)}, g \in PGC_{k=1}^{(1)}\}$. Since we know that $PGC_{k=i}^{(1)} \supset PGC_{k=1}^{(i)}$ (it is more general), we conclude that $PGC_{k=i+1}^{(1)} \supset PGC_{k=1}^{(i+1)}$. $\square$

## D    DATASETS

We empirically validated the proposed PGC-GNN on five commonly adopted graph classification benchmarks modeling bioinformatics problems: PTC (Helma et al., 2001), NCI1 (Wale et al., 2008), PROTEINS, (Borgwardt et al., 2005), D&D (Dobson & Doig, 2003) and ENZYMES (Borgwardt et al., 2005). The first two of them contains chemical compounds represented by their molecular graph, where each node is labeled with an atom type, and the edges represent bonds between them.

| Dataset | #Graphs | #Node | #Edge | Avg #Nodes/Graph | Avg.#Edges/Graph |
|---------|---------|-------|-------|------------------|------------------|
| **PTC** | 344 | 4915 | 10108 | 14.29 | 14.69 |
| **NCI1** | 4110 | 122747 | 265506 | 29.87 | 32.30 |
| **PROTEINS** | 1113 | 43471 | 162088 | 39.06 | 72.82 |
| **D&D** | 1178 | 334925 | 1686092 | 284.32 | 715.66 |
| **ENZYMES** | 600 | 19580 | 74564 | 32.63 | 124.27 |
| **COLLAB** | 5000 | 372474 | 24572158 | 74.50 | 4914.43 |
| **IMDB-B** | 1000 | 19773 | 193062 | 19.773 | 193.06 |
| **IMDB-M** | 600 | 19502 | 197806 | 13.00 | 131.87 |

Table 3: Datasets statistics.

PTC contains chemical compounds and the task is to predict their carcinogenicity for male rats. In NCI1 the graphs represent anti-cancer screens for cell lung cancer. The last three datasets, PROTEINS, D&D and ENZYMES, contain graphs that represent proteins. Each node corresponds to an amino acid and an edge connects two of them if they are less then 6Å apart. In particular ENZYMES, differently than the other considered datasets (that model binary classification problems) allows testing the model on multi-class classification over 6 classes. We additionally considered three large social graph datasets: COLLAB, IMDB-B, IMDB-M (Yanardag & Vishwanathan, 2015). In COLLAB each graph represents a collaboration network of a corresponding researcher with other researchers from three fields of physics. The task consists in predicting the physics field the researcher belongs to. IMDB-B and IMDB-M are composed of graphs derived from actor/actress who played in different movies on IMDB, together with the movie genre information. Each graph has a target that represents the movie genre. IMDB-B models a binary classification task, while IMDB-M contains graphs that belong to three different classes. Differently from the bioinformatics datasets, the nodes contained in the social datasets do not have any associated label. Relevant statistics about the datasets are reported in Table 3.

## E PGCN COMPUTATION COMPLEXITY EXAMPLE

Consider a dataset with $n_G$ graphs, and the 2-layers *GraphConv* defined with a message passing formulation in equations equation 6 and equation 7 (assuming $m_1 = m_2 = m$). Each *GraphConv* layer requires 3 matrix multiplications. The $\mathbf{AX}$ term in the first layer can be pre-computed since it remains the same over all training. Thus a 2-layer GCN performs $5 \cdot n_G$ matrix multiplications in the forward pass for each epoch (generally the size of $\mathbf{A}$ is different for each graph, but for the sake of discussion we can assume their dimension is comparable). Assuming 100 epochs for training, the total number of such multiplications is then $5 \cdot 100 \cdot n_G + 1$. If we now consider the PGC formulation with $k = 2$ in eq. equation 9 (that we recall is more expressive than 2 stacked *GraphConv* layers, as shown in Section 3.1), the number of matrix multiplications required for each graph is 6. However, the terms $\mathbf{AX}$ and $\mathbf{A}^2\mathbf{X}$ remain the same, for each graph, during all the training. They can thus be pre-computed and stored in memory. With this implementation, eq. equation 9 would require just 3 matrix multiplications, for a total number of matrix multiplications for 100 training epochs of $3 \cdot 100 \cdot n_G + 3$. While this does not modify the asymptotic complexity of PGC compared to message passing, it significantly improves the training times.

## F INITIAL NODE EMBEDDINGS

Some datasets that we used in the experiments, encode node labels (i.e., $\mathbf{X}$) by using a one-hot encoding. That makes the nodes representations very sparse. In preliminary experiments, we observed that such sparse representations negatively influence learning. In Table 4, we show how the use of a sparse node representation as input leads to have sparse input matrices $\mathbf{X}, \mathbf{AX}, \ldots, \mathbf{A}^k\mathbf{X}$. Specifically, in order to estimate the difference in terms of the sparsity degree with or without an initial PGC layer with $k = 1$, we computed the average ratio between the number of null entries (we round all the embedding values to the 4th decimal digit) and the total number of entries of the input matrices on the whole dataset for all the used bioinformatics datasets. We evaluated the sparsity of each PCG-layer block, considering the values of $k$ in the interval $[0, \ldots, 5]$. It is interesting to notice

| $input$ | PTC | NCI1 | PROTEINS | D&D | ENZYMES |
|---|---|---|---|---|---|
| $\mathbf{X}$ | 0.94 | 0.97 | 0.67 | 0.99 | 0.26 |
| $PGC_{k=1}(\mathbf{X})$ | 0 | 0 | 0 | 0 | 0 |
| $\mathbf{AX}$ | 0.93 | 0.96 | 0.45 | 0.95 | 0.13 |
| $\mathbf{A}\,PGC_{k=1}(\mathbf{X})$ | 0 | $3.67 \cdot 10^{-3}$ | $9.28 \cdot 10^{-5}$ | 0 | $1.87 \cdot 10^{-3}$ |
| $\mathbf{A}^2\mathbf{X}$ | 0.90 | 0.95 | 0.38 | 0.89 | 0.11 |
| $\mathbf{A}^2\,PGC_{k=1}(\mathbf{X})$ | 0 | $3.67 \cdot 10^{-3}$ | $9.28 \cdot 10^{-5}$ | 0 | $1.87 \cdot 10^{-3}$ |
| $\mathbf{A}^3\mathbf{X}$ | 0.89 | 0.95 | 0.36 | 0.86 | 0.10 |
| $\mathbf{A}^3\,PGC_{k=1}(\mathbf{X})$ | 0 | $3.67 \cdot 10^{-3}$ | $9.28 \cdot 10^{-5}$ | $2.39 \cdot 10^{-8}$ | $1.87 \cdot 10^{-3}$ |
| $\mathbf{A}^4\mathbf{X}$ | 0.88 | 0.94 | 0.35 | 0.87 | 0.10 |
| $\mathbf{A}^4\,PGC_{k=1}(\mathbf{X})$ | 0 | $3.67 \cdot 10^{-3}$ | $9.28 \cdot 10^{-5}$ | 0 | $1.87 \cdot 10^{-3}$ |
| $\mathbf{A}^5\mathbf{X}$ | 0.8 | 0.94 | 0.35 | 0.81 | 0.10 |
| $\mathbf{A}^5\,PGC_{k=1}(\mathbf{X})$ | 0 | $3.67 \cdot 10^{-3}$ | $9.28 \cdot 10^{-5}$ | $4.71 \cdot 10^{-8}$ | $1.87 \cdot 10^{-3}$ |

Table 4: Average ratio of the number of null entries over the total number of entries in the input components up to $k = 5$ without (top row) and with (bottom) the $PGC_{k=1}$ layer for the used datasets. Note that the value 0 corresponds to a dense matrix, while the value 1 to a null matrix.

that in all datasets the use of the initial PGC leads to a sparsity ratio near 0 (therefore the subsequent PGC-layer has in input dense embeddings). That is very useful, in particular for datasets like NCI1, PTC, and D&D, where the percentage of zeros in the labels representation is near 90%.

## G  PGCN IMPLEMENTATION DETAILS

We implemented the PGCN in PyTorch-Geometric (Fey & Lenssen, 2019). To reduce the *covariate shift* during training and to attenuate overfitting, we applied batch normalization and dropout on the output of each $\mathbf{y}_j$ layer. We used the *Negative Log Likelihood* loss, the *Adam* optimizer (Kingma & Ba, 2014), and the identity function for $\mathcal{T}$. For more details please check the publicly available code[1]. For our experiments, we adopted 2 types of machines, respectively equipped with:

- 2 x Intel(R) Xeon(R) CPU E5-2630L v3, 192GB of RAM and a Nvidia Tesla V100;
- 2 x Intel(R) Xeon(R) CPU E5-2650 v3, 160 GB of RAM and Nvidia T4.

## H  SPEED OF CONVERGENCE

Here, we discuss the results in terms of computation demand between a proposed PGCN and FGCNN (Navarin et al., 2020). We decided to compare these two models since they present a similar readout layer, therefore the comparison best highlights how the different methodology manage the number of considered k-hops, from the point of view of performance. In Table 5, we report the average time (over the ten folds) to perform a single epoch of training and to perform the classification with both method. In the evaluation we considered similar architectures using 3 layers for the FGCNN and $k = 3$ for PGCN. The other hyper-parameters were set with the aim to get almost the same number of parameters in both models, to ensure a fair comparison. The batch sizes used for this evaluation are the same selected by the PGCN model selection. The results show a significant advantage in using a PCG layer instead of the message passing based method exploited by FCGNN.
Concerning the speed of convergence of the two models, in Figure 2 we report the training curves for two representative datasets (D&D and NCI). In the x-axis we report the computational time in seconds, while in the y-axis we report the loss value. Both curves end after 200 training epochs. From the curves it can be seen that PGCN converges faster or with a similar pace than FCGNN.

---

[1]omitted for double-blind review.

| Dataset \ Model | Train | | Classification | |
|---|---|---|---|---|
| | **PGCN** | **FGCNN** | **PGCN** | **FGCNN** |
| D&D | 0.718±0.098 | 0.975±0.146 | 0.054±0.011 | 0.055±0.006 |
| ENZYMES | 0.164±0.015 | 0.247±0.032 | 0.011±0.001 | 0.016±0.002 |
| NCI1 | 0.883±0.119 | 1.568±0.263 | 0.052±0.005 | 0.089±0.011 |
| PROTEINS | 0.296±0.036 | 0.456±0.0599 | 0.024±0.003 | 0.027±0.004 |
| PTC | 0.084±0.009 | 0.139±0.016 | 0.006±0.002 | 0.009±0.003 |
| COLLAB | 1.507±0.175 | 2.048±0.378 | 0.137±0.014 | 0.109±0.014 |
| IMDB-B | 0.223±0.024 | 0.373±0.054 | 0.018±0.003 | 0.027±0.004 |
| IMDB-M | 0.326±0.044 | 0.554±0.087 | 0.022±0.003 | 0.034±0.005 |

Table 5: Time in second to perform a single training epoch ($2nd$ and $3rd$ column) and to perform classification ($4th$ and $5th$ column), using PGCN and FGCNN (Navarin et al., 2020), respectively.

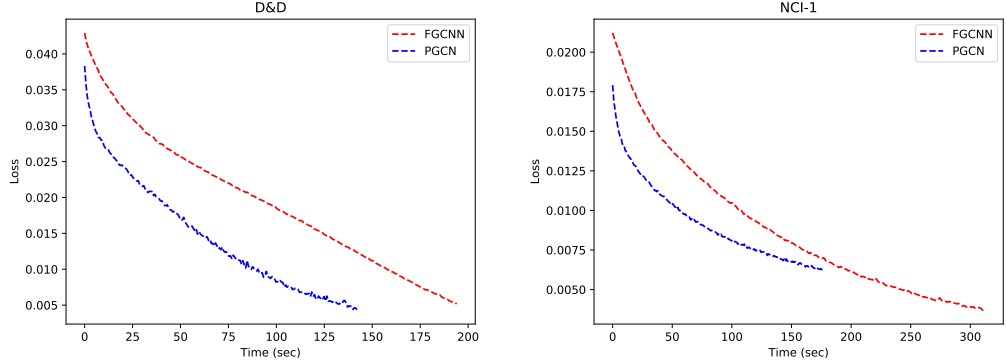

Figure 2: PGCN and FGCNN training curves for D&D and NCI-1 datasets.

## I   HYPER-PARAMETERS SELECTION

Due to the high computational time required to perform an extensive grid search, we decided to limit the number of values taken into account for each hyper-parameter, by performing preliminary tests to identify useful ranges of values.

The hyper-parameters of the model (number of hidden units, learning rate, weight decay, $k$) were selected by using a limited grid search, where the explored sets of values do change based on the considered dataset. Due to the high time requirements of performing an extensive grid search, we decided to limit the number of values taken into account for each hyper-parameter, by performing some preliminary tests. Preliminary tests showed that for the social network datasets, it is more convenient to use the Laplacian $\mathbf{L}$ as $\mathcal{T}(\mathbf{A})$. This behavior could be due to lack of label associated to nodes. In Table 6, we report the sets of hyper-parameters values used for model selection via grid search. As evaluation measure, we used the average accuracy computed over the 10-fold cross-validation on the validation sets, and we used the same set of selected hyper-parameters for each fold. For what concerns the selection of the epoch, it was performed for each fold independently based on the accuracy value on the validation set.

| Dataset/k | $m$ | learning rate | weight decay | drop out | batch size | k | readout(#layers [dims]) |
|---|---|---|---|---|---|---|---|
| **PTC** | $15, 30, 60$ | $10^{-3}, 5 \cdot 10^{-4}, 10^{-4}$ | $5 \cdot 10^{-3}, 5 \cdot 10^{-4}$ | $0.4, 0.6$ | $16, 32$ | $3, 4, 5, 6$ | $1\ [m/2], 2\ [m*2, m]$ |
| **NCI1** | $50, 100$ | $10^{-3}, 5 \cdot 10^{-4}$ | $5 \cdot 10^{-3}, 5 \cdot 10^{-4}$ | $0.3, 0.5$ | $16, 32$ | $3, 4, 5, 6$ | $1\ [m/2], 2\ [m*2, m]$ |
| **PROTEINS** | $25, 50$ | $10^{-3}, 5 \cdot 10^{-4}, 10^{-4}$ | $5 \cdot 10^{-3}, 5 \cdot 10^{-4}$ | $0.3, 0.5$ | $16, 32$ | $3, 4, 5, 6$ | $1\ [m/2], 2\ [m*2, m]$ |
| **D&D** | $50, 75$ | $5 \cdot 10^{-4}, 5 \cdot 10^{-5}$ | $5 \cdot 10^{-3}, 5 \cdot 10^{-4}$ | $0.3, 0.5$ | $16, 32$ | $3, 4, 5, 6$ | $1\ [m/2], 2\ [m*2, m]$ |
| **ENZYMES** | $50, 100$ | $10^{-3}, 10^{-4}$ | $5 \cdot 10^{-3}, 5 \cdot 10^{-4}$ | $0.3, 0.5$ | $16, 32$ | $3, 4, 5, 6$ | $1\ [m/2], 2\ [m*2, m]$ |
| **COLLAB** | $7, 15, 30$ | $10^{-3}, 5 \cdot 10^{-4}$ | $5 \cdot 10^{-3}, 5 \cdot 10^{-4}$ | $0, 0.5$ | $16, 32$ | $3, 5, 7, 9$ | $1\ [m/2], 2\ [m*2, m]$ |
| **IMDB-B** | $50, 75$ | $10^{-4}, 10^{-5}$ | $5 \cdot 10^{-4}, 5 \cdot 10^{-5}$ | $0, 0.5$ | $16, 32$ | $3, 5, 7, 9$ | $1\ [m/2], 2\ [m*2, m]$ |
| **IMDB-M** | $50, 75, 100$ | $10^{-4}, 5 \cdot 10^{-5}$ | $5 \cdot 10^{-3}, 5 \cdot 10^{-4}$ | $0, 0.5$ | $16, 32$ | $3, 5, 7, 9$ | $1\ [m/2], 2\ [m*2, m]$ |

Table 6: Sets of hyper-parameters values used for model selection via grid search.

| Model \ Dataset | PTC | NCI1 | PROTEINS | D&D | ENZYMES | COLLAB | IMDB-B | IMDB-M |
|---|---|---|---|---|---|---|---|---|
| GIN Xu et al. (2019) | 64.6 | 82.7 | 76.2 | - | - | 80.2 | 75.1 | 52.3 |
| | ±7.0 | ±1.7 | ±2.8 | - | - | ±1.9 | ±5.1 | ±2.8 |
| GFN Chen et al. (2019) | - | 82.77 | 76.46 | 78.78 | 70.17 | 81.50 | 73.00 | 51.80 |
| | - | ±1.49 | ±4.06 | ±3.49 | ±5.58 | ±2.42 | ±4.35 | ±5.16 |
| GCN Chen et al. (2019) | - | 83.65 | 75.65 | 79.12 | 69.50 | 81.72 | 73.30 | 51.20 |
| | - | ±1.69 | ±3.24 | ±3.07 | ±7.37 | ±1.64 | ±5.29 | ±5.13 |
| DIFFPOOL Ying et al. (2018) | - | - | 76.25 | 80.64 | 62.53 | 75.48 | - | - |
| PGCN | **74.0** | **84.40** | **79.20** | **82.54** | **78.17** | 76.96 | **77.88** | **52.97** |
| | **±2.81** | **±0.14** | **±0.81** | **±1.09** | **±0.68** | ±2.14 | **±3.11** | **±2.67** |

Table 7: PGCN accuracy comparison using different values of $k$. The validation policy is the same used in Xu et al. (2019); Chen et al. (2019); Ying et al. (2018). In Ying et al. (2018) the variance is not reported.

## J   EXPERIMENTAL RESULTS OMITTED IN THE RESULTS COMPARISON

As validation test methodology we decided to follow the method proposed in Errica et al. (2020), that in our opinion, turns out to be the fairest. For this reason, some results reported in the literature cannot be directly compared with the ones that we obtained.

Specifically, the results reported in Xu et al. (2019); Chen et al. (2019); Ying et al. (2018) are not considered in our experimental comparison since the model selection strategy is different from the one we adopted. Indeed the results reported r cannot be compared with the other results reported in Table 1 of the paper, because the authors state "*The hyper-parameters we tune for each dataset are [...] the number of epochs, i.e., a single epoch with the best cross-validation accuracy averaged over the 10 folds was selected.*". Similarly, for the result reported in Chen et al. (2019) for the GCN and the GFN models, the authors state "*We run the model for 100 epochs, and select the epoch in the same way as Xu et al. (2019), i.e., a single epoch with the best cross-validation accuracy averaged over the 10 folds is selected*". In both cases, the model selection strategy is clearly biased and different from the one we adopted. This makes the results not comparable.

Moreover, in Xu et al. (2019) the node descriptors are augmented with structural features. In GIN experiments the authors add a one-hot representation of the node degree. We decided to use a common setting for the chemical domain, where the nodes are labeled with a one-hot encoding of their atom type. The only exception is ENZYMES, where it is common to use 18 additional available features. Also in Ying et al. (2018) there is a similar problem since the authors add the degree and the clustering coefficient to each node feature vector. For the sake of completeness in Table 7 we report the results obtained by the proposed method following the same validation policy used in Xu et al. (2019); Chen et al. (2019); Ying et al. (2018). The table shows that the PGCN outperforms the methods proposed in the literature in almost all datasets.

