# OpenReview forum: "Polynomial Graph Convolutional Networks"
_ICLR.cc/2021/Conference — Reject_

### Official Review · AnonReviewer2 · 2020-10-16
**Owning more expressive ability but still lacking generalization**

**Rating:** 5
**Confidence:** 5

**Review:**

This paper analyzes the current graph convolution and proposes a novel polynomial graph convolutional network (PGCN). Under the framework of PGCN, several other graph convolutions can be seen as a special case.

Quality:
The paper is technically sound and maintains a high presentation quality.

Clarity:
The mathematical proof is clear and the whole paper is easy to follow.

Originality:
The paper is somewhat original but the proposed PGCN still lacks generalization, which will be pointed out in the following part.

Significance:
The significance is not very prominent.


Pros:
1. The paper proposes a novel graph convolution named with PGCN. Under the framework of PGCN, several other graph convolutions can be regarded as a special case.
2. The whole paper is described clearly and easy to follow.


Cons:
1. The core idea of PGCN mainly relies on the exponential power of different order applying on adjacent matrix A. It is easy to prove that by exponentiating the adjacency matrix to the K-th power, the node's receptive field can be extended to the K-th nearest neighboring nodes. However, this approach has been successfully adopted in different articles, as a result, the proposed PGCN is of limited originality without further analysis, either in theoretical aspect or technical aspect.
2. The mathematical proof in section 3.1 and section 3.2 shows that the stronger expressive ability comes from the less constraints of learnable parameters W. However, reducing the constraints may lead to the training process  more difficult than other graph convolution. To dispel this concern, the author needs to provide more detailed experimental results, especially the training convergence curves of different graph convolutions.
3. In terms of efficiency, the K-th order exponential method in equation (1) can improve the nodes' receptive field but brings heavy computational burden. In Table 3, All the evaluated dataset has a limited number of nodes in each graph. The maximum average number is 284.32 while the minimum average number is 13.00. The generalization of PGCN on large scale graph remains a serious concern. In Appendix E, the author has also noticed the computing efficiency and adopted a pre-computing strategy, which means the proposed PGCN is not an out-of-box model. In a word, although the proposed model shows stronger expression ability, the generalization and efficiency still remains a big concern.

---

> ### Author Response · Authors · 2020-11-17
> **Authors' Response**
>
> We thank the reviewer for writing this detailed review. In the following, we try to address the various highlighted cons. In the next few days we will upload the revised version of the paper.
>
> Comment:
> The core idea of PGCN mainly relies on the exponential power of different order applying on adjacent matrix A. It is easy to prove that by exponentiating the adjacency matrix to the K-th power, the node's receptive field can be extended to the K-th nearest neighboring nodes. However, this approach has been successfully adopted in different articles, as a result, the proposed PGCN is of limited originality without further analysis, either in theoretical aspect or technical aspect.
>
> Answer:
> In order to clarify the difference between our approach and other methods exploiting the exponentiations of the adjacency matrix recently proposed in the literature, in the revised version of the paper we will add a section where we report a complete review and comparison with these methods. However, we would like to stress that the majority of these methods are developed to perform nodes classification, while we propose an architecture that is designed to perform a different task (graph classification).
>
> --
>
> Comment:
> The mathematical proof in section 3.1 and section 3.2 shows that the stronger expressive ability comes from the less constraints of learnable parameters W. However, reducing the constraints may lead to the training process more difficult than other graph convolution. To dispel this concern, the author needs to provide more detailed experimental results, especially the training convergence curves of different graph convolutions.
>
> Answer:
> The experimental results show that the convergence of the PGCN is even faster than the common GC-based method in the literature. In order to clarify this point in the revised version of the paper we will add an appendix with some of the training curve plots for the various datasets. For sake of comparison, we will also report the convergence graph of the FGCNN that shares the same readout.
> We are sorry but we think we did not completely understand what the reviewer means with “training process more difficult”. In the case of  the reviewer referring to the complexity of the optimization, we point out that thanks to the constraint reduction introduced by the PGC layer, the optimization is less complex compared to stacked GC layers. If the reviewer refers to the efficacy of the training phase, the result obtained by PGCN in various datasets show the benefits of having a less constrained GC layer.
>
> --
>
> Comment:
> In terms of efficiency, the K-th order exponential method in equation (1) can improve the nodes' receptive field but brings heavy computational burden. In Table 3, All the evaluated dataset has a limited number of nodes in each graph. The maximum average number is 284.32 while the minimum average number is 13.00. The generalization of PGCN on large scale graph remains a serious concern. In Appendix E, the author has also noticed the computing efficiency and adopted a pre-computing strategy, which means the proposed PGCN is not an out-of-box model. In a word, although the proposed model shows stronger expression ability, the generalization and efficiency still remains a big concern.
>
> Answer:
> We respectfully disagree with the reviewer, since some of the datasets considered (e.g. COLLAB) have a significant amount of nodes per graph (and they are the most used dataset in the literature to benchmark GNNs). For what concerns the use of a pre-computing strategy, it is already used in other methods proposed in the literature (e.g. [1]), and in our opinion, this does not make the PGCN “not an out-of-box model”, since this step is automatically run when a new graph is given in input. To clarify this point and give more information about the computational efficiency of the PGCN,in the revised version of the papers, we will report the training time and a comparison with the training time required by the FGCNN.
>
> ---
>
> [1] Simplifying Graph Convolutional Networks

---

### Official Review · AnonReviewer4 · 2020-10-22
**Official Blind Review #4**

**Rating:** 5
**Confidence:** 4

**Review:**

Summary: The paper proposes Polynomial Graph Convolution (PGC), which enjoys a larger-than-one-hop receptive field within a single layer. This is done by first propagating information with a fixed (not learned) propagation matrix (e.g. adjacency matrix or graph Laplacian), and then projecting the information from different topological distances with a learned linear layer. PGC is shown to be theoretically more expressive than linearly stacking simple graph convolutions; experiments on several graph classification tasks show good performance.

Recommendation: Overall, I am slightly leaning to reject. The paper is interesting, but the theoretical contribution is rather straight-forward, and to warrant acceptance I would look for strong empirical performance together with an analysis of what is contributing to the gains. While some of the gains PGCN obtains look promising, others do not seem statistically significant; some results and baselines from related works are not compared against, and the effect of a strong node aggregation scheme is not discussed. Finally, it is unclear if the proposed method can be generalized to use edge types.

Main pros:
1. The paper is clear and easy to understand; even relatively small claims are backed up by proofs.
2. The relationship between linearly stacking k GraphConvs and using a single k-PGC layer is cleanly explained and motivates the work well. Decoupling receptive field size from network depth is a natural goal.
3. I appreciate the authors taking care to fairly evaluate their PGCN following [1]. This is crucial as many GNN works are iterating on metrics computed on the evaluation set despite it being available to the model selection procedure; as noted by authors, this includes the GIN results from [2].

Main cons:
1. I have several concerns about the results (Table 1):
- Best mean performance is bolded, but many of the gains (esp. PTC, PROTEINS, IMDB-B, IMDB-M) seem within noise levels due to high variance. Instead of saying "PGCN shows a slight improvement", it would be better to perform a statistical significance test, and clearly mark which improvements are significant (by bolding several joint-top values).
- Many results are taken from [1], but simple baselines of [1] that do not consider graph structure are omitted. These baselines outperform all GNNs in [1] for some benchmarks, e.g. getting 78.4 for D&D and 65.2 for ENZYMES (which is between the results of the best baseline and PGCN); to get a full picture these baselines should be included.
- [1] also shows that adding simple node features can make a big difference for IMDB-B and IMDB-M; for example, adding node degree features improves GIN on IMDB-B from 66.8 to 71.2. While PGCN is not using node degree features, for IMDB it's using the Graph Laplacian as T(A) (Appendix H), which implicitly encodes some degree information. As [1] already has results with node degree features included, it would be nice to see how PGCN compares.
- PGCN uses a powerful node aggregation scheme (Page 6). What aggregation methods are used in the baselines? What is the contribution of using this stronger aggregation scheme instead of a more standard weighted sum?
2. Section 3.3 and Appendix E mention that PGCN can be more efficient to train than standard GNNs because the propagation (i.e. multiplying node features by powers of T(A)) can be computed once. This only applies to the very first PGCN layer, and as authors state, before using a non-trivial (k > 1) PGC layer, one would typically use either 1-PGC (as done in the paper) or a learned linear projection, in order to avoid propagating highly sparse raw node features. It's therefore unclear if this training speedup can be really obtained in practice without sacrificing accuracy; one could use a random (not learned) linear projection as the initial "densification", but it's hard to say if that would work. Either way, if this speedup can really be obtained, it would be good to see supporting results (preferably on a large dataset where training time is indeed an issue).
3. Can PGC take edge types into account? I understand some of the benchmark datasets contain molecules, and thus have typed edges (single, double, triple); was that information utilized by PGCN? Considering edge types is straight-forward for GNNs that only perform a single learned propagation step, since the propagation matrix can be selected based on the edge type; it's unclear what to do for PGC which uses a fixed propagation matrix.

Other comments:
- The abstract and introduction say "interactions among GC parameters at different levels pose a bias on the flow of topological information". I'm assuming this refers to entanglement of weights as in Equation 8, but when reading the intro this got me confused, since there was no reference, and the claim did not seem obvious.
- Propagation in PGCN does not use any non-linearity or normalization, so it seems values can grow exponentially with k, especially on dense graphs. Was this observed to be a problem?
- "GraphConv is more expressive than GCN and GIN" - this is not fully true, since GCN as defined in the paper uses the Laplacian instead of the adjacency matrix.
- The analysis shows the relation between PGC and linearly stacking GraphConv's, but in practice one wouldn't linearly stack them, and rather use non-linearities in between. Can anything be said in that case?
- "they (...) consider shortest paths, (...) that choice limits significantly the expressiveness" - I'm not fully convinced: PGC cannot be stronger than the WL test, while using shortest path information allows to exceed that (since it makes it possible to tell apart two disconnected cycles from a single larger cycle).
- The paper compares only against GraphConv-like layers; it would be interesting to compare (empirically) to a larger class of GNNs, such as GAT [3], GGNN [4] or GNN-FiLM [5].

Typos and small formatting issues (did not influence my rating recommendation):
- Page 1: "faces the problem" -> "addresses the problem"
- Page 1: "being able to" -> "by being able to"
- Page 3: "start showing" -> "start by showing"
- Page 6: In equation for y_j, I believe j should start with 1
- Pages 6 & 14: "PCG" -> "PGC"


References:
- [1] A Fair Comparison of Graph Neural Networks for Graph Classification
- [2] How Powerful are Graph Neural Networks?
- [3] Graph Attention Networks
- [4] Gated Graph Sequence Neural Networks
- [5] GNN-FiLM: Graph Neural Networks with Feature-wise Linear Modulation

----------------------------------------------------------------------------------------------------

Comments after rebuttal:

I have reviewed the response from the authors, and I decided to keep my score. Although the paper is certainly interesting, for ICLR it is borderline. While I agree that many works in the literature choose to treat their network as a monolithic architecture, and thus do not explore the effects of different components, I would still encourage the authors to add an ablation of the readout method, as that would help assess the interaction between that and the choice of the graph propagation scheme.

---

> ### Author Response · Authors · 2020-11-17
> **Authors' Response part 1**
>
> We thank the reviewer for writing this detailed review. In the following, we respond to the various cons highlighted by the reviewer. In the next few days we will upload the revised version of the paper.
>
> Comment:
> I have several concerns about the results (Table 1):
> Best mean performance is bolded, but many of the gains (esp. PTC, PROTEINS, IMDB-B, IMDB-M) seem within noise levels due to high variance. Instead of saying "PGCN shows a slight improvement", it would be better to perform a statistical significance test, and clearly mark which improvements are significant (by bolding several joint-top values).
> Many results are taken from [1], but simple baselines of [1] that do not consider graph structure are omitted. These baselines outperform all GNNs in [1] for some benchmarks, e.g. getting 78.4 for D&D and 65.2 for ENZYMES (which is between the results of the best baseline and PGCN); to get a full picture these baselines should be included.
> [1] also shows that adding simple node features can make a big difference for IMDB-B and IMDB-M; for example, adding node degree features improves GIN on IMDB-B from 66.8 to 71.2. While PGCN is not using node degree features, for IMDB it's using the Graph Laplacian as T(A) (Appendix H), which implicitly encodes some degree information. As [1] already has results with node degree features included, it would be nice to see how PGCN compares.
> PGCN uses a powerful node aggregation scheme (Page 6). What aggregation methods are used in the baselines? What is the contribution of using this stronger aggregation scheme instead of a more standard weighted sum?
>
> Answer:
> Our experimental setting is the same used in [1] . Unfortunately we do not have access to the results of every single run for the baselines we report, but only to the final averages. Due to this issue, we are strongly limited in the statistical significance tests that can be computed. Note that this is a common issue in the field, and for this reason seldom statistical significance tests are computed. However we completely agree with the reviewer about the importance of performing statistical significance tests. Luckily we were able to compute the two-tailed Wilcoxon Signed-Ranks test between our proposed PGCN and competing methods. This test considers all the results for the different datasets at the same time. According to this test, PGCN performs significantly better (p-value<0.05) than PSCN, DGCNN, GIN, DIFFPOOL, and GraphSAGE. Meanwhile, for FGCNN and DGCNN, four datasets are not enough to conduct the test.
> We will report the results of the statistical significance test in the revised version of the paper.
>
> In the proposed comparison, we considered only the methods that exploit the topological information of the graph. The baseline in [1] is structure-agnostic and is used as a tool to understand the effectiveness of GNNs and extract useful insights.
> In the revised version of the paper, we will also report the results of this particular baseline. Note that the only dataset where the baseline outperforms the PGCN is PROTEINS, but with a significantly higher variance.
>
> For what concerns the use of node features in the social network dataset, we decided not to use them to evaluate the capability of the model when focusing on learning just from the graph structure.
> Using the normalized Laplacian (as we did for social datasets) is very different from using the degree as a node label, and indeed several approaches reported in the comparison use the Laplacian as well. Note that also the matrix A embeds the degree information, and thus the model can extract this information both when using T(A)=A and T(A)=L.
> We will include experiments also using the node degree as a label in a revision of the paper.
> For what concerns the PGC readout please note that we propose an architecture and not just a convolutional layer. The aggregation that we use is the same used in FGCNN, and it turns out to be less complex than the ones exploited by some other models in the comparison. For instance, the DIFFPOOL uses a particular pooling method, and the DGCNN uses the SortPooling layer followed by a 1D-convolutional layer followed by several dense layers (MLP).

---

> > ### Author Response · Authors · 2020-11-17
> > **Authors' Response part 2**
> >
> > Comment:
> > Section 3.3 and Appendix E mention that PGCN can be more efficient to train than standard GNNs because the propagation (i.e. multiplying node features by powers of T(A)) can be computed once. This only applies to the very first PGCN layer, and as authors state, before using a non-trivial (k > 1) PGC layer, one would typically use either 1-PGC (as done in the paper) or a learned linear projection, in order to avoid propagating highly sparse raw node features. It's therefore unclear if this training speedup can be really obtained in practice without sacrificing accuracy; one could use a random (not learned) linear projection as the initial "densification", but it's hard to say if that would work. Either way, if this speedup can really be obtained, it would be good to see supporting results (preferably on a large dataset where training time is indeed an issue).
> >
> > Answer:
> > Note that in the case of the first PGC layer with k=1 and a stacked Convolutional layer with k>1 it is still possible to precompute the power series of the adjacency matrix. We will update the paper to reflect this consideration.
> > Note that the choice of using a PGC layer instead of a random projection is due to the fact that using a random projection would have made it difficult to compare our results with other models considered in the comparison, since the same random projection of the input could have been applied to other models as well. In order to clarify this point we will add a  section in the revised version of the paper where we report some evaluation of the time required to train the PGCN compared with the one of other Graph Convolutional networks.
> >
> > --
> >
> > Comment:
> > Can PGC take edge types into account? I understand some of the benchmark datasets contain molecules, and thus have typed edges (single, double, triple); was that information utilized by PGCN? Considering edge types is straight-forward for GNNs that only perform a single learned propagation step, since the propagation matrix can be selected based on the edge type; it's unclear what to do for PGC which uses a fixed propagation matrix.
> >
> > Answer:
> > We focused our attention on graphs where edges do not have a type, similarly to the other models considered in the comparison that do not consider edge types. An extension of our approach allowing us to consider the type of edges is possible but not trivial and highly depends on the nature of the information related to the edges. For instance, in the specific case of molecules where the edge type denotes if the edge is single, double or triple, it is possible to represent this information using a weighted adjacency matrix. However, the extension to typed edges is outside the scope of our paper.
> > Note that the extension to typed edges is not trivial for the other models as well, since in many cases such an extension was published as a separate paper.

---

> > > ### Author Response · Authors · 2020-11-17
> > > **Authors' Response part 3**
> > >
> > > Comment:
> > > The abstract and introduction say "interactions among GC parameters at different levels pose a bias on the flow of topological information". I'm assuming this refers to entanglement of weights as in Equation 8, but when reading the intro this got me confused, since there was no reference, and the claim did not seem obvious.
> > > Propagation in PGCN does not use any non-linearity or normalization, so it seems values can grow exponentially with k, especially on dense graphs. Was this observed to be a problem?
> > > "GraphConv is more expressive than GCN and GIN" - this is not fully true, since GCN as defined in the paper uses the Laplacian instead of the adjacency matrix.
> > > The analysis shows the relation between PGC and linearly stacking GraphConv's, but in practice one wouldn't linearly stack them, and rather use non-linearities in between. Can anything be said in that case?
> > > "they (...) consider shortest paths, (...) that choice limits significantly the expressiveness" - I'm not fully convinced: PGC cannot be stronger than the WL test, while using shortest path information allows to exceed that (since it makes it possible to tell apart two disconnected cycles from a single larger cycle).
> > > The paper compares only against GraphConv-like layers; it would be interesting to compare (empirically) to a larger class of GNNs, such as GAT [3], GGNN [4] or GNN-FiLM [5].
> > >
> > >
> > > Answer:
> > > In the abstract when we discuss the bias on the flow of topological information posed by the interactions among non-linearly stacked graph convolution (GC) parameters at different levels we refer to the behavior highlighted (and proved) by theorem 2. Indeed in the theorem we prove that stacking 2 PGC layers with k=1 is less expressive than having a single PGC layer with k=2. Since we show in the paper (before the theorem) that the most common GC layers are just particular instances of the PGC layer with k=1, that means that stacking 2 or more (see corollary 2.1 in the appendix) GC layers is less expressive than a single PGC layer with k equal to the number of GC layers. Therefore it is straightforward to conclude that staking several GC layers poses a bias. We will clarify the sentence in the abstract.
> > >
> > > Actually, as we highlighted in appendix G, our model uses batch normalization, in particular after the aggregation layer. During the experiments we did not have any issue related with exponential growth of the nodes embeddings. Indeed it is important to note that without stacking several layers, and having independent weights for each k, the embedding values do not tend to significantly grow.
> > >
> > > Since the proposed PGC layer abstracts from the specific transformation of the adjacency matrix that is considered, the proof of theorem 2 is independent from the specific operator to represent the node connections. This is obvious from the proof (in Appendix).
> > >
> > > We agree that current state of the art models non-linearly stack GCN layers. In this paper we show that using a linear operator, and a more expressive architecture, helps the gradient flow and allows us to obtain better results than models explaining non-linearly stacked GCs (i.e. the state-of-the-art models we consider in our experimental comparison)
> > > When we discuss the expressiveness of the model proposed in [2] we refer to the use of a fixed transformation of the adjacency matrix. We completely agree with the reviewer and we will rewrite the discussion of the related paper.
> > > Note that we focus our work on the graph classification task, and the papers pointed out by the reviewer developed models for node classification, classification of sequences of graphs, or particular regression on the QM9 dataset (the exploits type of the edges). That poses a problem related to how to manage the readout part. Please note that the readout is a crucial part of our architecture. We focus our comparison on models that are developed and tested for the same purpose of the PGCN, and we decided to use the results available in the literature, adapting our validation method to perform a fair comparison.
> > >
> > > ---
> > >
> > > [1] A Fair Comparison of Graph Neural Networks for Graph Classification
> > > [2]On filter size in graph convolutional networks

---

### Official Review · AnonReviewer3 · 2020-10-28
**Interesting work, certain details missing**

**Rating:** 5
**Confidence:** 4

**Review:**

The article presents a novel framework for Graph Convolutional Neural Networks (GCNs). The method called  Polynomial Graph Convolution (PGC) is based on concatenating the powers of a transformed adjacent matrix in a given layer. The paper shows that various popular variants of GNNs can be expressed using the PGC framework.  Theoretical results presented show that PGC with higher degree is more expressive that deeper std. GNNs. Numerical results are presented on graph classification task that illustrate the performance of the method.

--------------
Strengths:
1. Paper presents a novel framework.
2. Numerical results look promising.
3. The method presents a more expressive method.

--------------
Weakness:
1. Novelty seems limited.
2. Node and edge task experiments are missing.
3. Certain claims need to be substantiated.

--------------
Details:
I have the following comments:
1. Novelty seems limited. It is unclear how the proposed method/framework is different from the Krylov subspace based work of (Luan et al. 2019) and also the Chebyshev polynomials approach of (Defferrard et al., 2016). Both these methods boil down to considering powers (polynomials) of the Laplacian, i.e., different topological distances in the model. Authors should clarify the key differences.

2. Experimental results section can be enhanced. Currently, the paper  considers only the graph classification task, where only global information suffices. GCN are more interesting in local, graph level tasks, in particular for prediction tasks on nodes and edges, e.g., link prediction, node and edge classification. It is unclear why these tasks were not considered. Moreover, methods of  (Kipf and Welling, 2016), (Luan et al. 2019) and (Defferrard et al., 2016), that are most relevant to this work are not considered for experimental comparisons.

3. The claim in Theorem 2, that PGC with k =2 is more expressive than two stacked PGC with k = 1, is du to that fact that the former has number of parameters higher than the latter. Is this correct? (Due to the structure of weight matrix W)

4.  There are few statements that need to be substantiated and explained. E.g.,
(a) It is claimed there is a bias in deeper GCNs, but no explanation is given.
(b) Similarly, PGC being more expressive (for the asme number of parameters) needs to be explained.

---

> ### Author Response · Authors · 2020-11-17
> **Authors' Response part 1**
>
> We thank the reviewer, whose suggestions helped us improve the clarity and quality of the paper. In the following we respond to the various weaknesses reported by the reviewer. In the next few days we will upload the revised version of the paper.
>
>
> Comment:
> Novelty seems limited. It is unclear how the proposed method/framework is different from the Krylov subspace based work of (Luan et al. 2019) and also the Chebyshev polynomials approach of (Defferrard et al., 2016). Both these methods boil down to considering powers (polynomials) of the Laplacian, i.e., different topological distances in the model. Authors should clarify the key differences.
>
> Answer:
> In order to give a complete answer to this weakness we will add a dedicated section in the revised version of the paper, where we highlight the difference between our proposed approach and the various architecture cited by the reviewer. Briefly we would like to highlight that in [2] the authors propose a model that stacks various layers and exploits nonlinear activation functions. Both these aspects make the gradient flow more complex compared to the PGCN. In [3] the model exploits the Chebyshev polynomials, and sums them over k. Note that the graph model proposed in [3] is an instance of the PGC, since the PGC is more general (as we show in section 3.1).
>
> --
>
> Comment:
> Experimental results section can be enhanced. Currently, the paper considers only the graph classification task, where only global information suffices. GCN are more interesting in local, graph level tasks, in particular for prediction tasks on nodes and edges, e.g., link prediction, node and edge classification. It is unclear why these tasks were not considered. Moreover, methods of (Kipf and Welling, 2016), (Luan et al. 2019) and (Defferrard et al., 2016), that are most relevant to this work are not considered for experimental comparisons.
>
> Answer:
> Please note that the architecture that we propose is specifically designed for graph classification. The similar models pointed out by the reviewer exploits the idea of using the power series of the adjacency matrix only to perform node classification. We exploited a similar base intuition and developed a model specific for the graph classification task. Thus, it is not possible to compare to those models since they are designed for a different learning task.
> We respectfully disagree with the reviewer’s statement that GCNs are more interesting in local tasks.
> Graph-level tasks such as graph classification/regression or generation are extremely important e.g. in chemoinformatics, where predicting properties of chemical compounds can significantly speed up the development of new drugs. For instance, we do not consider the JEDI Grand Challenge of virtual screening of a billion molecules for effectiveness against SARS-CoV-2 not interesting (https://www.covid19.jedi.group).
> Moreover, we used datasets that are largely used in all state-of-the-art papers in the field using an experimental validation very close to the one used in [1] (that provides an extensive validation via these datasets) so to have a solid placement of the performance of the proposed approach versus other SOTA competing approaches.
>
> --
>
> Comment:
> The claim in Theorem 2, that PGC with k =2 is more expressive than two stacked PGC with k = 1, is du to that fact that the former has number of parameters higher than the latter. Is this correct? (Due to the structure of weight matrix W)
>
> Answer:
> Theorem 2 proves that the PGCN is more expressive using the same amount of  parameters. The gain of expressiveness comes from the particular structure of the PGC that allows us to have a less constrained combination of the embedding computed for the various k values. The proof is in the appendix.

---

> > ### Author Response · Authors · 2020-11-17
> > **Authors' Response part 2**
> >
> > Comment:
> > There are few statements that need to be substantiated and explained. E.g., (a) It is claimed there is a bias in deeper GCNs, but no explanation is given. (b) Similarly, PGC being more expressive (for the asme number of parameters) needs to be explained.
> >
> > Answer:
> > Both these claims are related to theorem 2. When we discuss the bias on the flow of topological information posed by the interactions among non-linearly stacked graph convolution (GC) parameters at different levels we refer to the behavior highlighted (and proven) by theorem 2. Indeed in the theorem, we prove that stacking 2 PGC layers with k=1 is less expressive than having a single PGC layer with k=2. Since we show in the paper (before the theorem) that the most common GC layers are just particular instances of the PGC layer with k=1, that means that stacking 2 or more (see corollary 2.1 in the appendix) GC layers is less expressive than a single PGC layer with k equal to the number of GC layer. Therefore it is straightforward to conclude that staking several GC layers pose a bias. Note that Theorem 2 proves that the PGC with k=2 is more expressive than 2 staked PGC layers, in the case where the number of the parameters of the two configurations is the same.
> > To avoid confusion, we will clarify the sentence in the abstract.
> > For point (b), please see the previous answer.
> >
> >
> >
> > ---
> > [1] A Fair Comparison of Graph Neural Networks for Graph Classification
> > [2] Break the Ceiling: Stronger Multi-scale Deep Graph Convolutional Networks
> > [3] Convolutional Neural Networks on Graphs with Fast Localized Spectral Filtering

---

### Official Review · AnonReviewer1 · 2020-10-28
**Recommendation to Reject**

**Rating:** 4
**Confidence:** 4

**Review:**

##########################################################################

Summary:

This work proposes the Polynomial Graph Convolutional Networks (PGCNs), which is built upon the Polynomial Graph Convolution (PGC). The PGC is able to aggregate k-hop information in a single layer and comes with the hyper-parameter k. The PGCNs are composed of a PGC with k=1, followed by a PGC with a chosen k (usually > 1), and a complex readout layer using avg, max, and sum over all nodes. Theoretically, the proposed PGC has two major benefits as claimed: 1) Common graph convolution operators can be represented as special cases of the PGC; 2) A PGC with k = q (q > 1) is more expressive than linearly stacked q PGCs with k=1. The PGCNs are thus more general, expressive, and efficient than existing GNNs. Experimental studies are conducted on common graph classification benchmarks, showing the improved performances of the PGCNs.

##########################################################################

Reasons for score:

Overall, I vote for rejection. My major concerns lie in three aspects, as detailed in Cons below: 1) Some statements to motivate this work are not well explained and supported. 2) The proposed idea seems quite similar to several previous studies; 3) The experimental studies are not convincing to me in order to show the effectiveness of the proposed PGC.

##########################################################################

Pros:

1. The presentation is good and the proposed method is clearly described. I like the theoretical analysis in 3.2, which explains clearly how to build PGCNs with PGC.

2. This paper targets at two interesting problems in the literature: 1) designing more powerful graph convolution operator; 2) building GNNs with similar power as deeper GNNs. The deeper version of existing GNNs is usually not powerful as expected. As a result, building GNNs to achieve the expected power is an important topic.

3. The experimental details are provided in the appendix.

##########################################################################

Cons:

1. Some statements to motivate this work are not well explained and supported. In particular, in the abstract and section 1, the authors indicate that interactions among non-linearly stacked graph convolution (GC) parameters at different levels pose a bias on the flow of topological information. This statement is not well explained and supported with analysis or experimental results.

2. The proposed idea seems quite similar to several previous studies. The proposed PGC is quite similar (or equivalent, with minor differences) to concatenating outputs of linearly stacked GCs and then going through a linear transformation. First, such concatenation is proposed in [1]. Second, linearly stacking GCs, or equivalently using a polynomial powers of the adjacency matrix A (possibly with some transformations), is studies in multiple studies like [2,3,4]. I didn't capture the key differences between this work and these previous studies. Only [2] is discussed in the appendix. [4] is mentioned in the experiments, but no discussion in terms of method differences is provided.

[1] Xu et al. "Representation Learning on Graphs with Jumping Knowledge Networks", ICML 2018

[2] Wu et al. "Simplifying Graph Convolutional Networks", ICML 2019

[3] Liu et al. "Towards Deeper Graph Neural Networks", KDD 2020

[4] Chen et al. "Are Powerful Graph Neural Nets Necessary? A Dissection on Graph Classification", ICLR 2019 RLGM workshop

3. The experimental studies are not convincing to me in order to show the effectiveness of the proposed PGC. As mentioned in Summary, the PGCNs use a quite complex readout layer, which is different from all the GNNs in comparison. Ablation studies on this readout layer should be provided in order to show that the improvement is indeed from the PGC.

##########################################################################

Questions during rebuttal period:

Please address and clarify the cons above


##########################################################################

Comments after the rebuttal period:

I will keep my score. The authors' responses addresses my first concern. However, the differences from previous studies are still not significant to me. In addition, no matter the authors claim the proposal of a complete architecture or a new operator, ablation studies are necessary to backup the designing of each part.

---

> ### Author Response · Authors · 2020-11-17
> **Authors' Response part 1**
>
> We sincerely thank the reviewer for constructive criticism. In the following, we respond to the various cons raised by the reviewer. In the next few days we will upload the revised version of the paper.
>
> Comment:
> Some statements to motivate this work are not well explained and supported. In particular, in the abstract and section 1, the authors indicate that interactions among non-linearly stacked graph convolution (GC) parameters at different levels pose a bias on the flow of topological information. This statement is not well explained and supported with analysis or experimental results.
>
> Answer:
> In the abstract when we discuss the bias on the flow of topological information posed by the interactions among non-linearly stacked graph convolution (GC) parameters at different levels we refer to the behavior highlighted (and proved) by theorem 2. Indeed in the theorem we prove that stacking 2 PGC layers with k=1 is less expressive than having a single PGC layer with k=2. Since we show in the paper (before the theorem) that the most common GC layers are just particular instances of the PGC layer with k=1, that means that stacking 2 or more (see corollary 2.1 in the appendix) GC layers is less expressive than a single PGC layer with k equal to the number of GC layers. Therefore it is straightforward to conclude that staking several GC layers poses a bias. We will clarify the sentence in the abstract.
>
> Comment:
> The proposed idea seems quite similar to several previous studies. The proposed PGC is quite similar (or equivalent, with minor differences) to concatenating outputs of linearly stacked GCs and then going through a linear transformation. First, such concatenation is proposed in [1]. Second, linearly stacking GCs, or equivalently using a polynomial powers of the adjacency matrix A (possibly with some transformations), is studies in multiple studies like [2,3,4]. I didn't capture the key differences between this work and these previous studies. Only [2] is discussed in the appendix. [4] is mentioned in the experiments, but no discussion in terms of method differences is provided.
>
> Answer:
> We respectfully disagree and in order to better highlight the difference between the architecture we propose and the models proposed in literature, in the revised version of the paper we will add a section discussing these differences.
> Briefly, for what concerns the papers cited by the reviewer: in [1] the authors proposed to modify the common aggregation stage in such a way that, for each node, the model aggregates all the  intermediate representations computed in the previous GC-layers. Note that differently from PGCN, the model proposed in [1] exploits the message passing method introducing the bias in the flow of the topological information. We would like to stress the fact that a PGC layer of degree k is not equalivalent to concatenate the output of k stacked GC layers, even though the PGC layer can also learn to represent this particular architecture.
> In [2] the model proposed is basically a stack of linear GC operators. Note that in theorem 2 we prove that staking k GC layers is less expressive that using a single PGC layer of degree k.
> We thank the reviewer for reporting the work [3] of which we were not aware of when writing our manuscript, since [3] has been published only at the end of August 2020. We will discuss the differences with [3] in the revised version of the manuscript.
> In general, the majority of the models that concatenate polynomial powers of the adjacency matrix A are designed to perform node classification, while the proposed PGCN is developed to perform graph classification. In this regard, we want to point out that our proposal is not limited to a novel GC layer, but we propose a complete architecture to perform graph classification that performs significantly better (see our answer to AnonReviewer4) than state-of-the-art methods.

---

> > ### Author Response · Authors · 2020-11-17
> > **Authors' Response part 2**
> >
> > Comment:
> > The experimental studies are not convincing to me in order to show the effectiveness of the proposed PGC. As mentioned in Summary, the PGCNs use a quite complex readout layer, which is different from all the GNNs in comparison. Ablation studies on this readout layer should be provided in order to show that the improvement is indeed from the PGC.
> >
> > Answer:
> > We respectfully disagree since the readout of the proposed architecture is, in our opinion, quite simple. We use a simple MLP, experimenting with two alternatives: 1)  a single layer and 2) 3 fully connected readout layers. For what concerns the aggregation step, we use three simple element-wise operations (sum, mean, and max). Moreover, many of the architectures considered in the comparison use a more complex readout. For instance, the DIFFPOOL uses a pretty complex node pooling method,  while DGCNN uses the SortPooling layer followed by a 1D-convolutional layer followed by several dense layers (MLP). Finally, the aggregation method exploited in the PGCN is the same used by the FGCNN.
> > As mentioned before, we clarify that we propose a complete architecture to perform graph classification that performs significantly better (see our answer to AnonReviewer4) than state-of-the-art methods.

---

### Author Response · Authors · 2020-11-24
**Revised version of the paper**

We just submitted the revised version of the manuscript, where we addressed the issues reported by the reviewes concerning the originality of the model with respect to the GCN architectures in literature, the speed of convergenece, and the statistical significance of the experimental results.
We would like to stress that, even though the idea of exploiting the powers of the adjacency matrix is similar to other works in literature, our formulation differs enough to produce a *statistically significant* difference in the experimental results. By the way, also the differences among the works pointed out by reviewers are relatively small. Nonetheless, the works have been published anyway, based on improved (but not proven to be statistically significant) experimental results. Thus, we consider unfair to penalize our paper based mainly on this kind of argument.

---

### Decision · Program_Chairs · 2021-01-07
**Final Decision**

**Decision:**

Reject

**Comment:**

All four reviewers expressed significant concerns on this submission during review. None of them is willing to change their evaluations and supports this work during discussions. Thus a reject is recommended.